

**Light-absorbing black carbon and brown carbon components of smoke aerosol from**
**DSCOVR EPIC measurements over North America and Central Africa**
Myungje Choi[1,2], Alexei Lyapustin[2], Gregory L. Schuster[3], Sujung Go[1,2], Yujie Wang[1,2], Sergey
Korkin[1,2], Ralph Kahn[2,4], Jeffrey S. Reid[5], Edward J. Hyer[5], Thomas F. Eck[1,2], Mian Chin[2], David
J. Diner[6], Olga Kalashnikova[6], Oleg Dubovik[7], Jhoon Kim[8], Hans Moosmüller[9]
[1]Goddard Earth Sciences Technology and Research (GESTAR) II, University of Maryland
Baltimore County, Baltimore, MD, USA
[2]NASA Goddard Space Flight Center, Greenbelt, MD, USA
[3]NASA Langley Research Center, Hampton, VA, USA
[4]Laboratory for Atmospheric and Space Physics, The University of Colorado Boulder, Boulder,
CO, USA
[5]US Naval Research Laboratory, Monterey, CA, USA
[6]Jet Propulsion Laboratory, California Institute of Technology, Pasadena, CA, USA
[7]Laboratoire d'Optique Atmosphérique, Université de Lille-1, CNRS, Villeneuve d'Ascq, France
[8]Department of Atmospheric Sciences, Yonsei University, Seoul, Republic of Korea
[9]Laboratory for Aerosol Science, Spectroscopy, and Optics, Desert Research Institute, Reno, NV,
USA
Correspondence to: Myungje Choi (myungje.choi@nasa.gov)





## Abstract

Wildfires and agricultural burning generate seemingly increasing smoke aerosol emissions, impacting societal and natural ecosystems. To understand smoke's effects on climate and public health, we analyzed the spatiotemporal distribution of smoke aerosols, focusing on two major light-absorbing components, black carbon (BC) and brown carbon (BrC) aerosols. Using NASA's Earth Polychromatic Imaging Camera (EPIC) instrument aboard the NOAA's Deep Space Climate Observatory (DSCOVR) spacecraft, we inferred BC and BrC volume fractions and particle mass concentrations based on spectral absorption provided by the Multi-Angle Implementation of Atmospheric Correction (MAIAC) algorithm with 1-2 hours temporal resolution and ~10 km spatial resolution over North America and Central Africa. Our analyses of regional smoke properties reveal distinct characteristics for aerosol optical depth (AOD) at 443 nm, spectral single scattering albedo (SSA), aerosol layer height (ALH), and BC and BrC amounts. Smoke cases in North America show extremely high AOD up to 6, with elevated ALH (6-7 km) and significant BrC components up to 250 mg/m$^2$ along the transport paths, whereas the smoke aerosols in Central Africa exhibited stronger light absorption (i.e., lower SSA) and lower AOD, resulting in higher BC mass concentrations and similar BrC mass concentrations than the cases in North America. Seasonal burning source locations in Central Africa following the seasonal shift of Inter Tropical Convergence Zone and diurnal variations in smoke amounts were also captured. Comparison of retrieved $AOD_{443}$, $SSA_{443}$, $SSA_{680}$, and ALH with collocated AERONET and CALIOP measurements shows agreement with *rmse* of 0.2, 0.03-0.04, 0.02-0.04, and 0.8-1.3 km, respectively. Analysis of spatiotemporally average reveals distinct geographical characteristics in smoke properties closely linked to burning types and meteorological conditions. Forest wildfires over western North America generated smoke with small BC volume fraction of 0.011 and high ALH with large variability (2.2 ± 1.2 km), whereas smoke from wildfires and agricultural burning over Mexico region shows more absorption and low ALH. Smoke from savanna fires over Central Africa has the most absorption with high BC volume fraction (0.015) and low ALH with small variation (1.8 ± 0.6 km) among the analyzed regions. Tropical forest smoke was less absorbing and had a high variance in ALH. We also quantify the estimation uncertainties related to the assumptions of BC and BrC refractive indices. The MAIAC EPIC smoke properties with BC and BrC volume and mass fractions and assessment of layer height provide observational constraints for radiative forcing modeling and air quality and health studies.

**Keywords**: EPIC, light absorbing smoke aerosol component, BC, BrC



## 1. Introduction

Natural and anthropogenic fires affect and shape nearly every terrestrial vegetated ecosystem on the planet (Pausas and Keeley, 2009; Bond and Keeley, 2005), and their emissions have long been known to affect the global atmospheric composition and radiative budget (Hobbs et al., 1997; Seiler and Crutzen, 1980). Recent climate changes and anthropogenic activities have affected wildfire and agricultural fire occurrence in many regions (Liu et al., 2010; Dennison et al., 2014). Global monitoring of atmospheric smoke aerosol chemical, optical, and microphysical properties is important to quantify the impacts of increasing biomass burning on climate and air quality. However, the current understanding of smoke aerosol radiative forcing is still insufficient due to its high spatiotemporal variability in combination with the dynamic nature of smoke and variability of its physical and optical properties (IPCC, 2023).

One characteristic that distinguishes smoke particle components from other components is light absorption. Absorbing particle components converting incident electromagnetic energy into thermal energy results in heating of both the particles and the ambient surrounding atmosphere. Aerosol light absorption greatly affects direct radiative forcing and atmospheric stability and convections (IPCC, 2023; Bellouin et al., 2005; Yu et al., 2002). Smoke particles emitted from biomass burning typically contain two major light-absorbing carbonaceous components: black carbon (BC) and brown carbon (BrC). The proportions of these light-absorbing components and their mixing ratios determine the spectral absorption characteristics (e.g., Jacobson, 2001; Chakrabarty et al., 2023).

BC is a byproduct of the incomplete combustion of carbonaceous materials. There is no specific chemical makeup of BC and depending on measurement techniques it is also called soot, elemental carbon, or light-absorbing carbon (Reid et al., 2005a; Moosmüller et al., 2009; Andreae and Gelencsér, 2006). BC is visibly black, resulting in a high and spectrally invariant imaginary refractive index (~0.79) across UV-visible wavelengths (Bond and Bergstrom, 2006). During combustion, tiny BC spherules are aggregated with each other and grow by absorbing surrounding gas-phase molecules into large particles with a complex, generally fractal-like morphology (Moosmüller et al., 2009). Emitted atmospheric BC particles are generally hydrophobic (Petters et al., 2009), but can quickly evolve to hydrophilic if they acquire water-soluble coatings upon emission or during atmospheric aging (Tritscher et al., 2011). Atmospheric aging processes change BC's physical and chemical particle structure (Corbin et al., 2023; Bhandari et al., 2019; Sengupta et al., 2020), as well as optical properties (Gyawali et al., 2017; Kleinman et al., 2020; Reid et al., 2005b). Particle evolutions combine with the high spatial and temporal variability of the sources to make the net radiative effects of these particles highly uncertain (Bond et al., 2013; IPCC, 2023; Chakrabarty et al., 2023).

The largest carbonaceous aerosol component directly emitted from biomass burning is organic carbon (OC; e.g., Andreae and Merlet, 2001; Andreae, 2019 and references therein). This study defines the OC with significant light absorbing property in the tropospheric solar spectrum as brown carbon (BrC; e.g., Laskin et al., 2015). BrC exhibits spectral variability, absorbing more ultraviolet (UV) and short visible light than long visible light, resulting in a reddish or brownish





appearance. Its imaginary refractive index varies spectrally, with generally higher values at shorter
(i.e., UV) wavelengths and decreasing toward longer, visible and infrared (IR) wavelengths
(Kirchstetter et al., 2004). BrC emission and the chemical processes responsible for BrC formation
are complex and not yet fully understood. Some studies suggest BrC consists primarily of water-
soluble organic carbon compounds and humic-like substances (Sun et al., 2007; Phillips and Smith,
2014; Hoffer et al., 2006) whereas others suggest that non-polar compounds can absorb more light
than polar compounds, especially in the UV and short-wavelength visible (Sengupta et al., 2018).
BrC compounds can be released from smoldering biomass burning or formed through secondary
organic aerosol processes in the atmosphere (Chakrabarty et al., 2010; Laskin et al., 2015). BC
coated with non-absorbing organic and inorganic may exhibit a similar wavelength dependence of
absorption, with higher values at shorter wavelengths (Wang et al., 2016). This similarity makes
it challenging to differentiate between BrC and coated BC based on spectral absorption alone.
Therefore, our "BrC" results may include contributions from coated BC.

According to the latest Intergovernmental Panel on Climate Change (IPCC) report (IPCC,

2023), the present day global effective radiative forcing of black carbon from fossil fuel and biofuel
is estimated at 0.107 W m$^{-2}$ with a 5-95% uncertainty range of $-0.202$ to 0.417 W m$^{-2}$, with
respect to the pre-industrial time of 1750. In contrast, primary organic aerosols from fossil fuel
and biofuel, related to OC, exhibit a cooling effect of $-0.209$ W m$^{-2}$, with an uncertainty range of
$-0.439$ to $-0.021$ W m$^{-2}$. Although BrC is not directly considered in this assessment, its radiative
forcing is partially accounted for within primary organic aerosol, biomass burning, or secondary
organic aerosols in some global aerosol models. Combining ground-based measurements and
chemical transport modeling, Jo et al. (2016) attributed non-BC absorption to BrC and estimated
BrC fraction as 21% of the global mean surface OC concentration, significantly impacting ozone
photochemistry by altering the UV radiation field. Zhang et al. (2020) estimated that the global
BrC direct radiative effect is 0.10 W m$^{-2}$, suggesting that BrC can heat the tropical mid and upper
troposphere more than BC. Still, much uncertainty remains about BrC due to limited measurements
and the complex processes involved, challenging accurate estimates of its radiative impact on
climate (Liu et al., 2020).

Intensive *in situ* measurements have been instrumental in identifying the composition-

related spectral light-absorption properties of smoke plumes, as summarized in Bond and
Bergstrom (2006), Andreae and Gelencsér (2006), Moosmüller et al. (2009), and Samset et al.
(2018). These measurements have enabled remote sensing techniques to differentiate between
various light-absorbing components in smoke plumes. For example, the Aerosol Robotic Network
(AERONET) sunphotometers routinely provide aerosol optical and microphysical properties,
including spectral refractive indices from many sites worldwide (Holben et al., 1998; Dubovik and
King, 2000). Using AERONET inversion data, Schuster et al. (2016) inferred aerosol components
over smoke- and dust-dominated regions by matching AERONET spectral refractive index to
mixtures of components with different assumed optical properties. Specific absorbing components
were assumed as inclusions: BC and BrC for smoke and iron oxides of hematite and goethite for
dust aerosols. Wang et al. (2013) and Choi et al. (2020) applied a similar approach to East Asia





sites. The synergy between visible/near-IR AERONET measurement and UV/visible multifilter
rotating shadowband radiometer (MFRSR) measurements confirmed the sensitivity of spectral
absorption consistent with a BrC component (Mok et al., 2016, 2018).

Inferring aerosol composition from satellites is more challenging than from ground-based
remote sensing due to the need to account for the surface contribution to the top-of-atmosphere
signal, and the much greater range of conditions that space-borne instrument samples. Retrieving
aerosol absorptions using multi-spectral bands in near UV wavelengths has been applied to
instruments such as the Total Ozone Mapping Spectrometer (TOMS) and the Ozone Monitoring
Instrument (OMI), which have data records spanning decades, as well as more recently launched
instruments like the TROPOspheric Monitoring Instrument (TROPOMI) and Earth Polychromatic
Imaging Camera (EPIC; Torres et al., 1998, 2007, 2013, 2020; Ahn et al., 2021). The fraction of
retrieved single scattering albedo (SSA) within the expected error, defined as a fraction within
±0.03 from AERONET SSA, is approximately 50%, based on long-term and global validation
across these sensors (Ahn et al., 2021; Torres et al., 2020).

The Generalized Retrieval of Aerosol and Surface Properties (GRASP) algorithm
(Dubovik et al., 2011, 2014) utilizes the multi-angle, multi-channel, and both radiometric and
polarimetric measurements from the POLarization and Directionality of the Earth's Reflectances
(POLDER) instruments. With increased information incorporated by a multi-pixel multi-temporal
smoothness constraint, the GRASP algorithm retrieves aerosol optical depth (AOD), particle size
information, and absorption, showing robust agreement with global AERONET measurements
(Chen et al., 2020). Recent improvement of the GRASP algorithm included the direct estimation
of aerosol chemical composition concentrations without the need for intermediate steps such as
retrieving refractive indices and particle size distributions (Li et al., 2019, 2020). The Multi-angle
Imaging SpectroRadiometer (MISR) research algorithm also accounts for black-smoke and brown-
smoke aerosol models (Limbacher et al., 2022), analogous to the BC and BrC components in this
study, and is utilized to analyze fractional AODs along transport paths (Junghenn Noyes et al.,
2020a, b, 2022). Still, it is worth noting that POLDER and MISR measurements are limited to
visible and near-infrared (NIR) channels and do not include ultraviolet (UV) channels, where
spectral absorption due to BC and in particular BrC is more pronounced.

The EPIC sensor aboard the Deep Space Climate Observatory (DSCOVR) spacecraft has
provided UV-near IR measurements of Earth since 2015 (Marshak et al., 2018). Recent studies by
(Lyapustin et al., 2021b) have utilized the Multi-Angle Implementation of Atmospheric Correction
(MAIAC) processing of EPIC measurements to derive AOD and spectral absorption. It enables
inferring aerosol chemical compositional differences, such as BC and BrC in smoke aerosol
plumes and iron oxides (e.g., hematite and goethite) in dust aerosol plumes. DSCOVR's orbit
around the Lagrange-1 point, where the spacecraft remains stably positioned between the sun and
Earth, allows for global monitoring multiple times per day during the daylight time with a temporal
resolution of 1-2 hours. In our study, we used EPIC measurements to infer BC and BrC volume
fractions and mass concentrations in smoke plumes and identified distinct smoke properties over



North America and Central Africa. The estimation of iron oxides in dust aerosols using the EPIC
MAIAC product was addressed in Go et al. (2022).
The structure of the paper is as follows. Section 2 introduces the EPIC MAIAC smoke
aerosol retrieval algorithm and describes the methodology for inferring BC and BrC volume
fractions and mass concentrations. It also includes descriptions of study regions and of AERONET
and CALIOP validation datasets. In Section 3, we analyzed individual smoke cases over North
America and Central Africa, and provided validation of AOD, spectral SSA, and aerosol layer
height (ALH). Additionally, time-integrated regional properties, including BrC/BC ratios, and
uncertainty estimates based on different inclusion assumptions are discussed. Finally, Section 4
offers summary and concluding remarks.
**2. Data and methods**
**2.1 MAIAC EPIC processing algorithm**

EPIC measurements cover the entire sunlit hemisphere of Earth with ten narrowband
spectral channels from 317.5 to 779.5 nm. The spatial resolution of EPIC is ~8-16 km at nadir,
degrading toward the edge of the image. MAIAC EPIC algorithm grids and processes L1B data at
10 km resolution providing an oversampling. DSCOVR's Lagrange point 1 orbit between the Earth
and the Sun (~1.5 million kilometers) enables global multi-temporal daytime measurements, with
10−12 observations in boreal summer and 6-7 observations in winter at mid-latitudes and little
seasonal change in tropical latitudes. Detailed information on EPIC measurements can be found
in Marshak et al. (2018). Following the MAIAC Moderate Resolution Imaging Spectroradiometer
(MODIS) algorithm (Lyapustin et al., 2018), the standard MAIAC processing offers cloud
detection, atmospheric correction, and AOD with regionally specified background aerosol models
("background AOD"; Lyapustin et al., 2021a). In addition, a newly developed absorbing smoke or
dust aerosol retrieval process was applied to both land and ocean pixels. Smoke/dust detection and
separation are based on various tests including UV aerosol index and spectral AOD shape. As
EPIC band configuration does not allow to distinguish between smoke and dust aerosols, the dust
retrievals are only performed over pre-defined dust regions whereas smoke retrievals are
performed elsewhere globally (Lyapustin et al., 2021b).
The full algorithm description is given elsewhere (Lyapustin et al., in preparation); here
we provide a very brief overview to facilitate understanding of our results. The novel version 3
(v3) MAIAC algorithm represents spectral aerosol absorption with two parameters, the imaginary
refractive index at 680 nm ($k_0$) and spectral absorption exponent (SAE), using a conventional
power-law expression, $k_\lambda = k_0 (\lambda/\lambda_0)^{-SAE}$ where $\lambda_0 = 680$ nm. The real refractive index is
assumed to have a spectrally invariant value of 1.51 (Lyapustin et al., 2021b). The particle log-
normal volume size distribution is defined as $\frac{d\,V(r)}{d\ln(r)} = \sum_{i=1}^{2} \frac{C_{Vi}}{\sqrt{2\pi}\sigma_i} e^{-\frac{1}{2}\left(\frac{\ln(r) - \ln(r_{v,i})}{\sigma_i}\right)^2}$ , where i
indicates each mode (fine and coarse), r is the particle radius, $r_{v,i}$ is the volume mean radius, $\sigma_i$ is



the geometric standard deviation, $c_{v,i}$ is the volumetric concentration. For smoke aerosols, we
assumed fine mode volume mean radius (0.14 µm) and geometric standard deviation (0.4 µm),
coarse mode volumetric mean radius (2.8 µm) and geometric standard deviation (0.6 µm). In
MAIAC v3, the Levenberg-Marquardt nonlinear optimal fitting algorithm (Levenberg, 1944;
Marquardt, 1963) is used to simultaneously retrieve four parameters {$AOD_{443}$, $k_0$, SAE, ALH} by
matching EPIC measurements at UV to NIR wavelengths, including oxygen A and B bands. The
algorithm uses pre-computed look-up tables (LUTs) covering the full range of expected variability
of the above parameters. The maximum value of AOD at 443 nm in the algorithm is set to 6.
Vertically, the aerosol is modeled by a single 2 km-thick aerosol layer placed at different altitudes
in the atmosphere, and the reported ALH is defined as the midpoint height of the layer. To avoid
systemic biases in absorption, this retrieval is performed over detected absorbing smoke/dust
pixels when the retrieved AOD, based on the background aerosol model with fixed regional
properties, is greater than 0.4. Note that although smoke retrievals are limited with "background
$AOD_{443} > 0.4$" the retrieved smoke $AOD_{443}$ could be lower than 0.4 due to different assumption
of microphysical properties and simultaneous retrieval of spectral absorption and ALH.
**2.2 MAIAC smoke composition inference**
Given a very different spectral absorption of BC (high and spectrally fairly flat) and BrC
(low and strongly increasing towards UV), the retrieved spectral absorption can be used to derive
fractions of absorbing components. We assume that smoke aerosols consist of a non-absorbing
host and two absorbing species, BC and BrC, with internal mixing based on Maxwell Garnett
effective medium approximation (MG-EMA) (Bohren and Huffman, 1998; Schuster et al., 2005,
2016). The MG-EMA is suitable for characterizing smoke particles and is computationally
efficient (Garnett, 1904; Bohren and Huffman, 1998; Schuster et al., 2005; Markel, 2016a, b). For
that reason, it is widely used for inferring aerosol compositions from ground-based or satellite-
based remote sensing (Li et al., 2019; Schuster et al., 2005, 2016; Choi et al., 2020; Go et al., 2022).
Studies showed that different mixing rules, such as Bruggeman approximation or volume
averaging, yields similar results to the MG-EMA for inferring smoke components (Schuster et al.,
2016; Li et al., 2019; and references therein). The non-absorbing host (or medium) represents a
mixture of non-absorbing or low-absorbing components in smoke, such as non-absorbing OC,
sulfate, nitrate, and/or ammonium. Although there are various ranges of refractive indices for both
BC and BrC based on literature and experiments, this study assumes fixed refractive index to
estimate their fractions from the limited information of the retrieved optical properties. The BC
refractive index assumes Bond and Bergstrom (2006)'s suggestion of spectrally flat with a real
part (n) of 1.95 and an imaginary part (k) of 0.79 for the visible spectrum (i.e., $400 - 700$ nm).
Spectral dependence of k for BrC is based on Kirchstetter et al. (2004), whereas a constant real
part of 1.54 was assumed based on Li et al. (2019). For a spectrally flat and non-absorbing host
we assume n=1.51, consistent with the smoke aerosol model in the MAIAC EPIC algorithm, and
$k=10^{-9}$ based on Kalashnikova et al. (2018). Table 1 summarizes the spectral refractive indices of



BC, BrC and host. Please note that a sensitivity test for different assumptions regarding BC and
BrC imaginary refractive indices affecting their volume fractions is detailed in Sec 3.5.
Table 1. Spectral refractive indices of smoke aerosol components at EPIC wavelengths.

|                  | BC   |       | BrC  |       | host |           |
| ---------------- | ---- | ----- | ---- | ----- | ---- | --------- |
| Wavelengths (nm) | n    | k     | n    | k     | n    | k         |
| 340              | 1.95 | 0.790 | 1.54 | 0.187 | 1.51 | $10^{-9}$ |
| 388              | 1.95 | 0.790 | 1.54 | 0.125 | 1.51 | $10^{-9}$ |
| 443              | 1.95 | 0.790 | 1.54 | 0.070 | 1.51 | $10^{-9}$ |
| 680              | 1.95 | 0.790 | 1.54 | 0.003 | 1.51 | $10^{-9}$ |

The MG-EMA equation for smoke aerosol mixtures, as described in Bohren and Huffman
(1998) and Schuster et al., (2005), is presented below.
$$\epsilon_m = \epsilon_h \left[ 1 + \frac{3\left( f_{BC}\frac{\epsilon_{BC} - \epsilon_h}{\epsilon_{BC} + 2\epsilon_h} + f_{BrC}\frac{\epsilon_{BrC} - \epsilon_h}{\epsilon_{BrC} + 2\epsilon_h}\right)}{1 - f_{BC}\frac{\epsilon_{BC} - \epsilon_h}{\epsilon_{BC} + 2\epsilon_h} - f_{BrC}\frac{\epsilon_{BrC} - \epsilon_h}{\epsilon_{BrC} + 2\epsilon_h}} \right]$$

Here, $\epsilon_m$, $\epsilon_h$, $\epsilon_{BC}$, and $\epsilon_{BrC}$ represent the complex dielectric functions of the mixture, host,
BC, and BrC, respectively, and $f_{BC}$ and $f_{BrC}$ denote the volume fractions of BC and BrC,
respectively. Note that identical BC and BrC components are assumed for both fine and coarse
modes. Throughout plume evolution, different processes such as oxidation, hydration, deposition
of volatile organics onto existing particles, or new particle formation, may lead to larger particle
sizes. Consequently, the fine-mode and coarse-mode components in smoke aerosols could exhibit
differences. Schuster et al. (2016) also accounted for different component combinations between
fine and coarse modes, considering dust particles for the coarse mode. It should be noted that
biomass burning aerosols are strongly dominated by the fine mode component, with typically only
a minor coarse mode AOD. However, the MAIAC EPIC processing relies on a static particle size
distribution, and dynamic separation of fine and coarse modes is challenging with limited
measurement information.
The refractive indices of the mixture can be determined using the following equations:
$$n = \sqrt{\frac{\sqrt{\epsilon_r^2 + \epsilon_i^2} + \epsilon_r}{2}},$$

$$k = \sqrt{\frac{\sqrt{\epsilon_r^2 + \epsilon_i^2} - \epsilon_r}{2}},$$



where $\epsilon_r$ and $\epsilon_i$ represent the real and imaginary parts of the mixture dielectric function $\epsilon_m$. Given
fixed spectral refractive indices of the host and inclusions (BC and BrC), the mixture refractive
indices are determined by the volume fractions of two inclusions ($f_{BC}$ and $f_{BrC}$).
Subsequently, we utilized the Levenberg-Marquardt nonlinear least-square fitting method
(Levenberg, 1944; Marquardt, 1963; Press et al., 2007) to derive the volume fractions of inclusions
by comparing inferred and calculated refractive indices with the MG-EMA. Retrieved $k_0$ and SAE
were converted into spectral imaginary refractive indices ($k_\lambda$ for $\lambda$ of 340, 380, 443, and 680 nm)
and matched with theoretical values of a mixture to find solutions for $f_{BC}$ and $f_{BrC}$.
Fig. 1 illustrates the derivable BC, BrC, and host volume fractions for assumed ranges of
$k_0$ (0.001−0.016) and SAE (0.1−4) in the MAIAC EPIC algorithm. Available $f_{BC}$, $f_{BrC}$, and $f_{host}$
ranges are from 0 to 0.025, 0.994, and 0.998, respectively, where $f_{host} = 1 - f_{BC} - f_{BrC}$. The
maximum $f_{BC}$ of 0.025 can be found in the condition of maximum $k_0$ of 0.016. A high $f_{BrC}$ near
one can be retrieved when both $k_0$ and SAE are high. The host volume fraction ($f_{host}$) shows an
opposite tendency to $f_{BrC}$ and is low when both $k_0$ and SAE are high. Conversion from retrieved
$k_0$ and SAE to volume fractions follows the presented distributions.
It should be mentioned that the upper limit of $k_0$=0.016 was found empirically based on
limited EPIC regional processing, and then confirmed by the global processing of EPIC data.
However, this limit may be increased in the future based on detailed analysis of EPIC retrievals,
in particular because AERONET inversion retrievals often show higher values, for example in
Central and southern Africa savanna burning region (Eck et al., 2003).

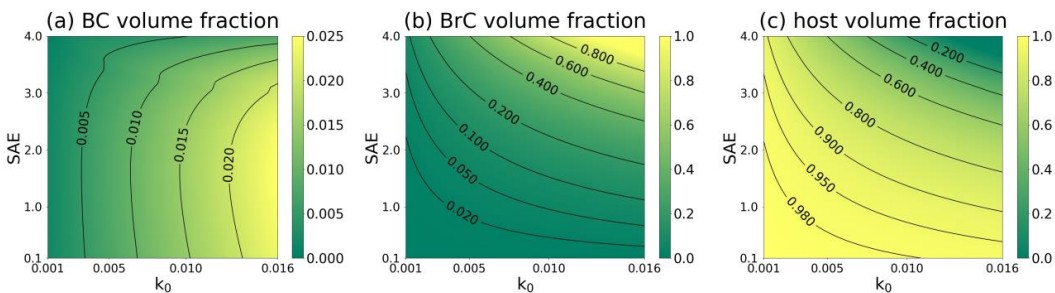

Figure 1. The range of volume fractions for (a) BC, (b) BrC, and (c) host across different values
of $k_0$ and SAE.

The inferred volume fractions of BC and BrC can be converted to column-integrated
volume concentrations as,
$$C_V = C_{Vf} + C_{Vc} = \frac{AOD_f}{h_f} + \frac{AOD_c}{h_c},$$

$$AOD_f = AOD \cdot \left(\frac{C_{Vf}}{C_{Vf}+C_{Vc}}\right),$$

$$AOD_c = AOD \cdot \left(\frac{C_{Vc}}{C_{Vf}+C_{Vc}}\right),$$



where $C_V$ is the column-integrated volume concentration with a unit of $\mu m^3/\mu m^2$, the subscripts f
and c indicate fine-mode and coarse-mode, respectively. Despite the regional dependence of $\frac{C_{Vc}}{C_{Vf}}$
in the 4D-retrieval algorithm for smoke, we assume a static $\frac{C_{Vc}}{C_{Vf}}$ of 0.7 for BC and BrC processing
to maintain consistency and reduce regional discrepancies arising from the ratio. Hygroscopicity
was neglected by using a static AOD per volume concentration regardless of relative humidity.
Given the size distribution and n, $h_f$ of 8.43 $\mu m^2/\mu m^3$ is fine mode $AOD_{443}$ per unit volume
concentration ($\mu m^3/\mu m^2$) and $h_c$ of 0.72 $\mu m^2/\mu m^3$ is coarse mode $AOD_{443}$ per unit volume
concentration, as calculated based on Mie theory in the MAIC EPIC smoke model (Lyapustin et
al., 2021b). Given the complex refractive indices, size distribution with fine-mode or coarse-mode
only, and non-sphericity, the h values, representing total column AOD per unit volume
concentration, are computed using the DLS (sphere and spheroid) model (Dubovik et al., 2006) at
volume concentration of 1 $\mu m^3/\mu m^2$. $h_f$ and $h_c$ are computed separately for the fine and coarse
modes within the MAIAC look-up table generation package and can be used to assess mass
extinction efficiency (MEE) with assumption of particle density. The column-integrated mass
concentration of the chemical component is calculated as $C_{M,i} = C_V \cdot f_i \cdot \rho_i$, where i indicates
inclusions (BC and BrC) and $\rho$ is mass concentration per unit volume. We use $\rho_{BC}$ of 1.8 $g/cm^3$
and $\rho_{BrC}$ of 1.2 $g/cm^3$ following previous studies (Bond and Bergstrom, 2006; Turpin and Lim,
2001; Schuster et al., 2016; Li et al., 2020).

### 2.3 Study regions

We selected two major regions where smoke aerosols are dominant but exhibit different
characteristics: North America (170°W-50°W and 13°N-80°N) and Central Africa (8°E-42°E and
17°S-5°N). To avoid potential interference from dust aerosols on smoke analysis, we excluded the
Sahel region bounding the Sahara Desert from this study. The selected smoke aerosol analysis
regions, along with detected fire counts from the Visible Infrared Imaging Radiometer Suite
(VIIRS) instrument onboard the Suomi National Polar-orbiting Partnership (SNPP) satellite in
2018, are presented in Fig 2. This study focused on the entire year of 2018, a year marked by one
of highest monthly average AOD during the summer over North America (Eck et al., 2023). The
EPIC dataset exhibited no temporal gaps, and ample AERONET and CALIOP data were
accessible. Additionally, we included a single case study from 2017 to complement our analysis
over North America.

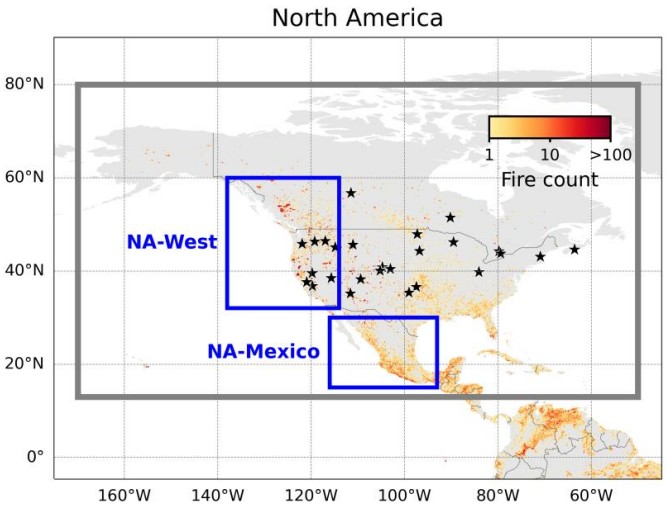

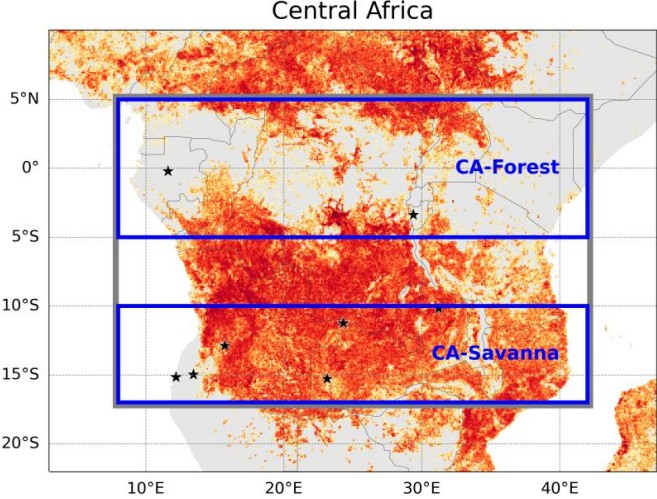

Figure 2. Cumulative fire detection counts from VIIRS within a 0.1° by 0.1° longitude-latitude
grid in 2018 over North America and Central Africa. The study regions are denoted by grey
rectangles, and AERONET locations are marked with blue stars. Subregions including western
("NA-West") and Mexico ("NA-Mexico") in North America, as well as tropical forest ("CA-
Forest") and savanna ("CA-Savanna) regions in Central Africa are denoted by blue rectangles.
**2.4 AERONET**
In order to evaluate the EPIC-retrieved AOD and spectral absorption, we utilized the
Version 3 Level 2.0 AERONET Inversion dataset (Holben et al., 1998; Dubovik and King, 2000;





Giles et al., 2019; Sinyuk et al., 2020). The EPIC-retrieved $AOD_{443}$, $SSA_{443}$ and $SSA_{680}$ were
compared with the AERONET counterpart derived from direct and sky radiance measurements.
The AERONET measurements of spectral AOD have accuracy of ~0.01 to 0.02 at optical airmass
of one with higher uncertainty in the UV (Eck et al., 1999). The AERONET retrieved SSA at 440
nm have uncertainty of ~0.03 at AOD(440)=0.4 with smaller uncertainties at larger AOD,
decreasing to ~0.015 at AOD(440)=1.3 for biomass burning aerosols at the Mongu, Zambia site
(Sinyuk et al., 2020). Spatiotemporal collocation between AERONET and EPIC measurements
was conducted as follows: (1) averaging AERONET AOD within a ±30-min range and averaging
SSA within a ±3-hour range from the EPIC measurement time, and (2) averaging EPIC $5 \times 5$ pixels
(~$50 \times 50$ km$^2$) collocated with the AERONET sites and limited to cosines of solar zenith angle
and view zenith angle above 0.45 (i.e., solar zenith angle & view zenith angle < 63.3°). The EPIC
pixels were spatially averaged when at least 50% of EPIC smoke products are valid in the spatial
window. AERONET retrievals with extinction Ångström exponent between 440 and 675 nm
greater than 0.4 were selected to avoid possible dust contamination. SSA validation was conducted
only when AERONET AOD at 440 nm was greater than 0.6. The AERONET sites with at least
five measurements available were considered. Consequently, a total of 28 and 7 AERONET sites
were chosen over North America and Central Africa, respectively (see Fig 2).
**2.5 CALIOP**
The Cloud-Aerosol Lidar with Orthogonal Polarization (CALIOP) onboard Cloud-Aerosol
Lidar and Infrared Pathfinder Satellite Observations (CALIPSO) satellite has provided global
measurements of aerosol vertical distribution. We collected profiles of total attenuated backscatter
coefficients at 532 nm ($\beta$, unit of km$^{-1}$sr$^{-1}$) from the CALIPSO Lidar Level 2 Aerosol Profile
version 4.51 dataset ("CAL_LID_L2_05kmAPro-Standard-V4-51") in 2018. Subsequently, we
calculated backscatter-weighted aerosol layer height using the formula $ALH_{CALIOP} = \frac{\sum \beta z}{\sum \beta}$, where
z represents the height of each layer. This definition is widely employed for validating aerosol
layer height using CALIOP (Go et al., 2020; Xu et al., 2019). The $ALH_{CALIOP}$ data within a ±30
min from EPIC acquisitions were spatially averaged within MAIAC EPIC grid. We used the same
cutoff threshold for the Sun and view zenith angle as above. To mitigate ALH uncertainty for weak
aerosol cases, the ALH comparison was conducted when CALIOP AOD at 532 nm exceeded 0.6.
**3. Results**
**3.1 Analysis of individual cases**
**3.1.1 North America**
Western North America stands out as one of the most active wildfire regions globally. For
our analysis, we selected an intense wildfire and associated smoke aerosol event occurring on July
18, 2017, at 20:52:19 Coordinated Universal Time (UTC) over western Canada in Fig 3. Note that



all other analyses in this study are for 2018 except for this case. Utilizing the VIIRS/SNPP Thermal
Anomalies/Fire (Schroeder and Giglio, 2018), visualized as red dots within the true-color images,
we identified wildfires in British Columbia. The true-color image and retrieved smoke particle
properties illustrate the eastward transport of the smoke plume. Specifically, pixels near the
wildfires (region "A" in Fig 3) exhibited $AOD_{443}$ nearing ~4-6, alongside an $SSA_{443}$ of ~0.93.
Pixels approximately 50~100 km from the sources (region "B"), show decreased $AOD_{443}$ (~2) and
less absorption ($SSA_{443}$ of ~0.96). Notably, the contrast in SSA is more pronounced at 388 nm
than at 680 nm (not shown). Absorption changes within this distance are related to the aging
process. Freshly emitted particles from wildfires exist in various mixing states and undergo
multiple processes, such as coagulation, condensation/evaporation, oxidation, and secondary
aerosol particles formed from chemical production (Reid et al., 2005a, b; Liu et al., 2020). Smoke
aerosol mixtures become less absorbing in the UV and shortwave visible wavelengths when
transported from sources through these aging processes, consistent with findings from other *in-*
*situ* and remote sensing measurement studies (Junghenn Noyes et al., 2020a, b; Kleinman et al.,
2020). The increased $SSA_{443}$ from 0.93 to 0.96 (from region "A" to "B") corresponds to a decrease
in the BrC fraction from 0.3 to 0.1. Aerosol plumes over Alberta, farther downwind to the east
(region "C"), exhibited a) high $AOD_{443}$ values (1-3), b) $SSA_{443}$ of ~0.92-0.94, c) increased BC
volume fraction up to 0.01; and d) a similar BrC volume fraction (about 0.3 at the plume center)
for pixels close to the fire sources. The eastern part of the plumes was located farther away from
the source and could have undergone more extensive aging. Smoke aerosol near sources was
located close to the surface (ALH above sea level of ~1 km) and was elevated to about 5-6 km in
the downwind area. It is important to consider that the fires could also undergo various stages of
combustion intensity over time, which could also be a factor in BC and BrC production. The
observed differences in ALH suggest that possibly some of these fires were more intense earlier,
leading to the lofting of the plume to 5-6 km. Subsequently, the intensity may have decreased,
resulting in a lower ALH as the plume transitioned to a more smoldering phase. This scenario,
particularly applicable to long plume lengths, implies that fire intensity and the relative combustion
fraction (flaming/smoldering) likely varied over the course of several hours during the transport
of such a long plume distance.



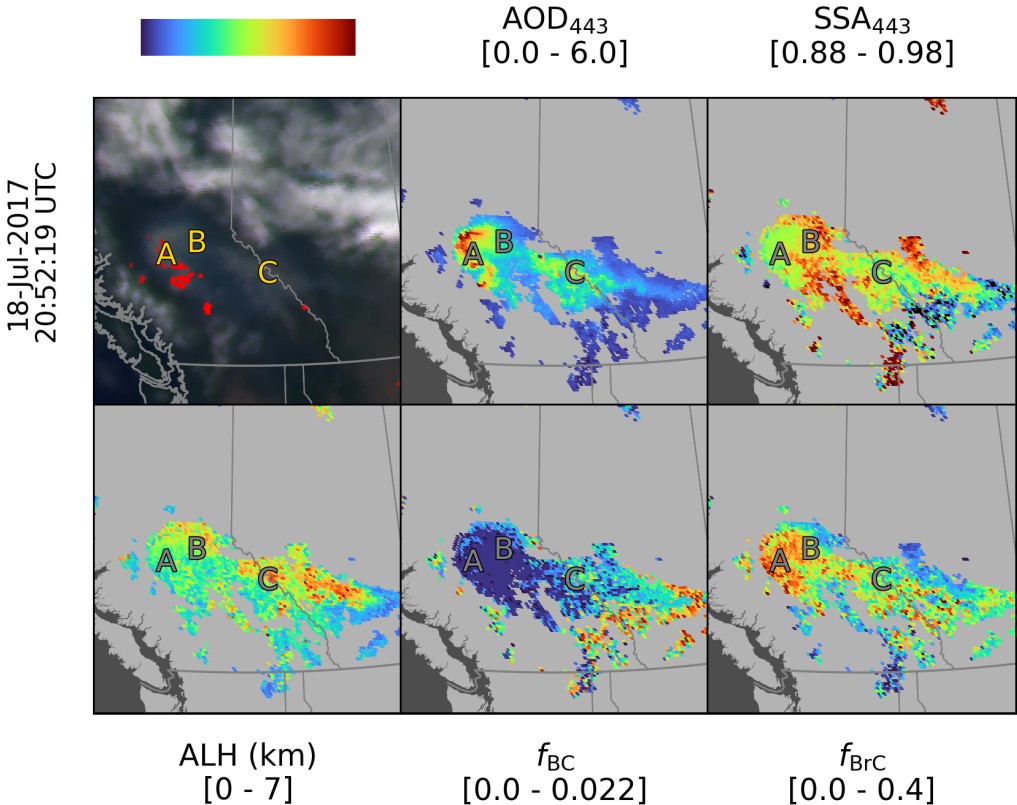

Figure 3. Illustration of EPIC smoke aerosol optical properties over western North America on
July 18, 2017. Red dots in the first left panel are VIIRS/NPP thermal anomaly hotspots. The
underlying image and analyses in subsequent panels correspond to EPIC true color and MAIAC
EPIC retrievals (AOD$^{443}$, SSA$_{443}$, and ALH) with inferred BC and BrC volume fractions. The color
bar scale is indicated at the top of each panel.
Continental-scale smoke aerosol episodes in August 2018, derived from the analysis of
Lyapustin et al. (2021b), are depicted in Fig 4. On August 13 (top panels), smoke aerosol plumes
along the west coast of North America, near the detected wildfire sources, exhibit high AOD of
nearly 3-4 and SSA$_{443}$ of 0.93 in the plume center. Surrounding pixels of the plume generally show
lower AOD and higher SSA than the pixels interior to the plume. Subsequently, westerly
transported plumes with increased AOD (up to ~6) and ALH (~6-7 km) were detected on August
16 and 17. Corresponding BC and BrC fractions ranged from 0.005 to 0.01 and 0.2 to 0.3 (not
shown), with column mass concentrations reaching 15 mg/m$^2$ and 250 mg/m$^2$, respectively.

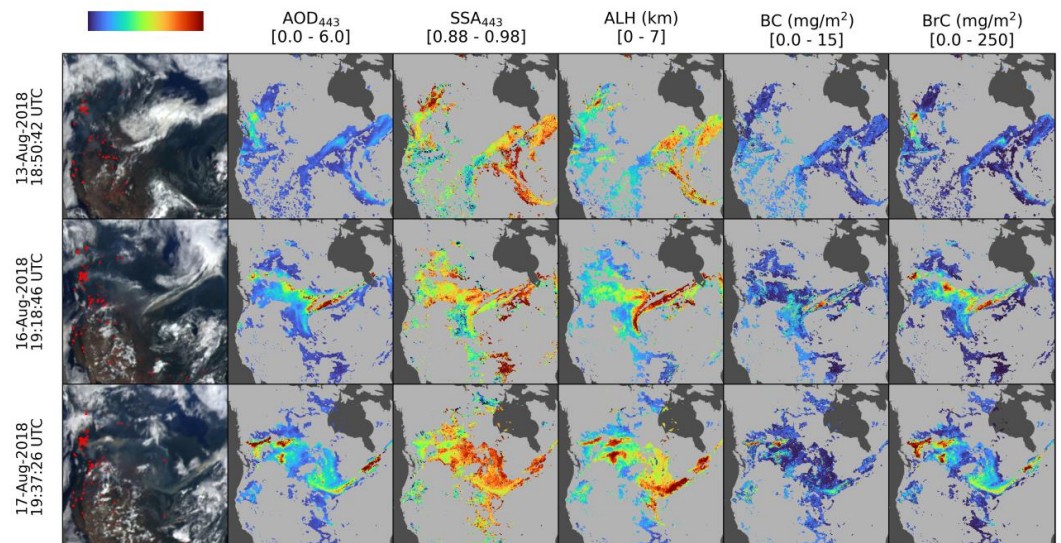

Figure 4. Illustration of smoke aerosol optical properties (AOD$_{443}$, SSA$_{443}$, ALH, and BC and BrC mass concentrations) over North America on August 13, 16, and 17, 2018. The color bar scale is indicated at the top of each panel.

EPIC can effectively monitor the change of smoke optical properties during transport at high temporal cadence. Meridional averages of AOD$_{443}$, SSA$_{443}$, ALH, and BC and BrC mass concentrations over the period from August 13 to 17, 2018 are represented as Hovmöller diagrams in Fig 5. Plume evolution is clearly captured, with a temporal resolution of 1-2 hours, from initial smoke aerosol emission over western North America, to subsequent transport toward the east with an increased ALH from ~1 km to 6-7 km, and eventually to dispersion.

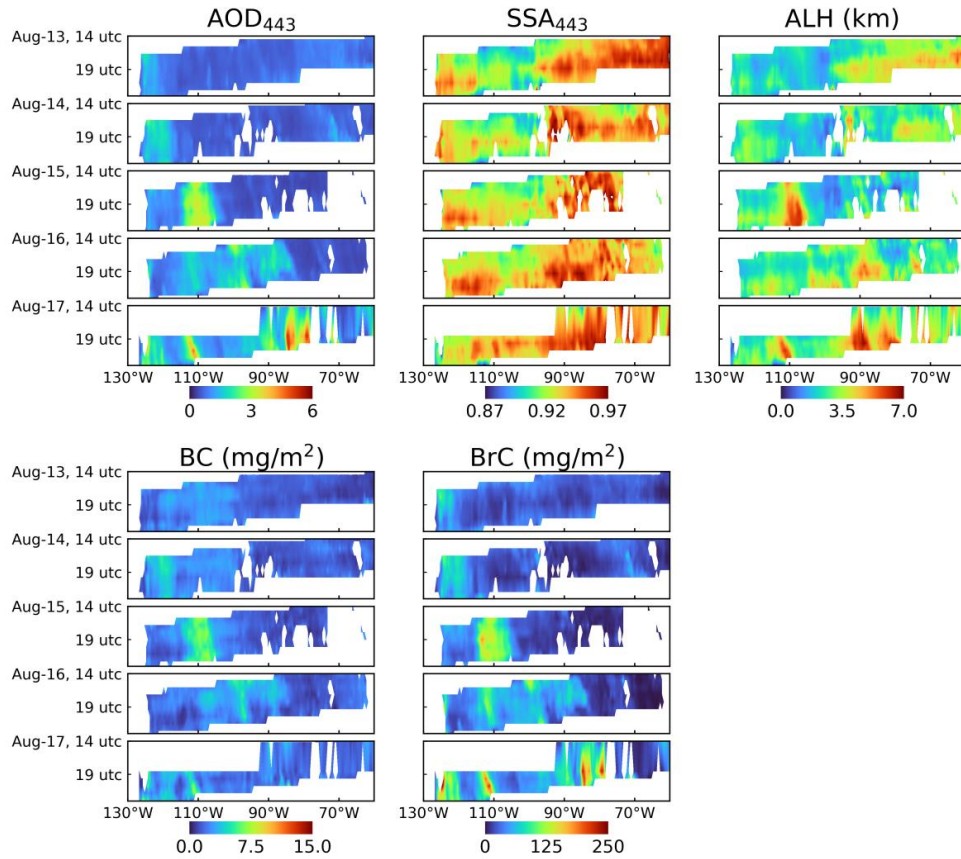

Figure 5. Hovmöller diagrams of AOD$_{443}$, SSA$_{443}$, ALH, and BC and BrC volume fractions over
North America (130-60°W, 25-53°N, 0.5° longitudinal interval) from August 13 to 17, 2018. Gaps
in the data are due to low AOD or meteorological clouds.
**3.1.2 Central Africa**

Biomass burning over Central Africa generates smoke aerosols with distinct optical
properties. Long-term AERONET measurements over Southern Africa savanna regions indicate
the strongest absorption among global smoke regions, with SSA values at 440 and 680 nm of 0.87
and 0.86, respectively (Dubovik et al., 2002; Giles et al., 2012; Sayer et al., 2014). The biomass
burning emission pattern in Africa follows a clearly defined seasonal cycle, influenced by
precipitation linked to the seasonal movement of the Inter-Tropical Convergence Zone (ITCZ)
(Swap et al., 2003). There exists a strong temporal cycle of SSA as well, with the lowest SSA
values in June due to savanna burning, and increasing through October as more forested areas burn



(Eck et al., 2013). And yet, particle size distributions tend to remain unchanged (Reid et al., 2005b;
Sayer et al., 2014). This makes the region an ideal test environment for absorption retrievals. We
selected four cases (June 8, July 26, August 14, and September 19, 2018) to illustrate the seasonal
changes in smoke regions from northeast to southwest; these align closely with the climatological
patterns detected by other ground-based and satellite measurements (Eck et al., 2013; Duncan et
al., 2003). The detected fires were subcontinent-wide (Fig 6) and generated smoke with AOD
reaching up to ~2. The general particle properties were consistent across the four cases. The light
absorption, reaching as low as ~0.84 $SSA_{443}$, was notably stronger than in the cases over North
America. The ALH of pixels with high AOD remained relatively constant at 2-3 km. High BC
concentrations (e.g., > 5 mg/m$^2$) were prevalent over detected fire locations despite relatively
lower AOD condition (e.g., $AOD_{443}$ < 2) than in the cases over North America, where similar BC
concentrations were observed from the pixels with $AOD_{443}$ > ~3.

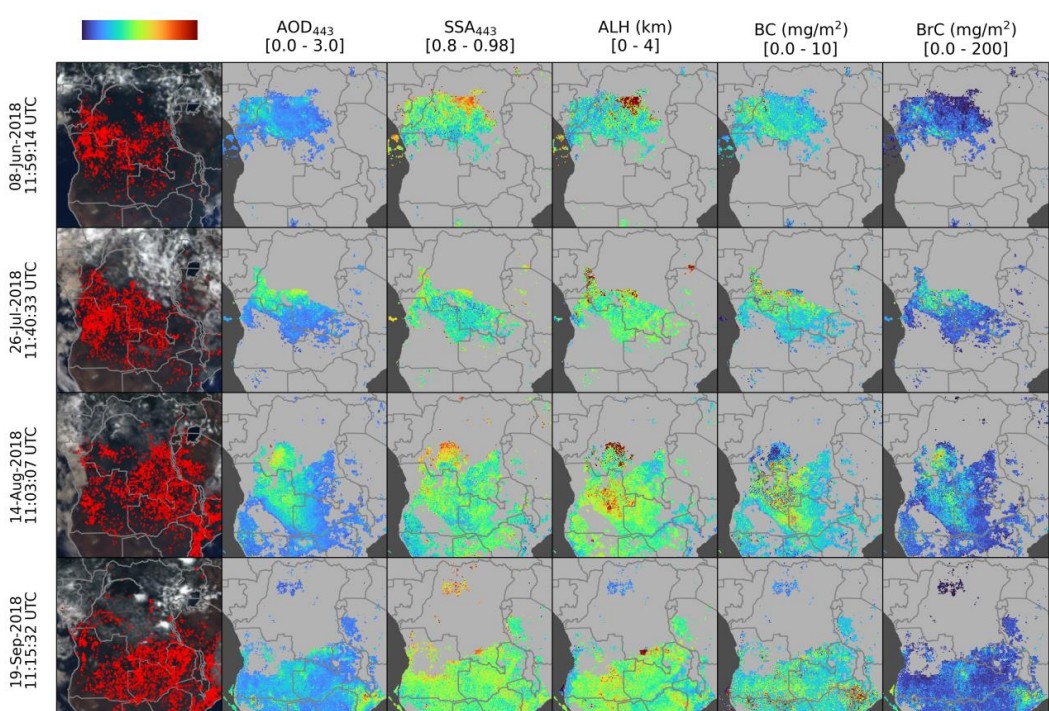

Figure 6. Illustration of smoke optical properties ($AOD_{443}$, $SSA_{443}$, ALH, and BC and BrC mass
concentrations) over Central Africa on June 8, July 26, August 14, and September 19, 2018.
The measurements taken over five consecutive days from August 13-17 over the Central
Africa study region detected weaker zonal smoke plume transport with less dynamic changes in
particle properties (Fig 7) compared to the North America cases (Fig 5). The relatively low ALH




of 2-3 km indicates that smoke aerosol mostly concentrated within the boundary layer and was
less influenced by strong jets at higher altitudes. AOD was slightly enhanced during early morning
and late afternoon by ~10-20% over 20-25°E region. The afternoon pattern is consistent with long-
term AERONET measurements shown in Eck et al. (2003), whereas the morning pattern should
be further analyzed. From SEVIRI measurements, the peak of active fires is most frequently
detected around noon (Wooster et al., 2021). Eck et al. (2003) concluded that elevated air
temperatures, reduced relative humidity, and heightened wind speeds during the midday and
afternoon periods often lead to more intense and rapidly spreading fires.

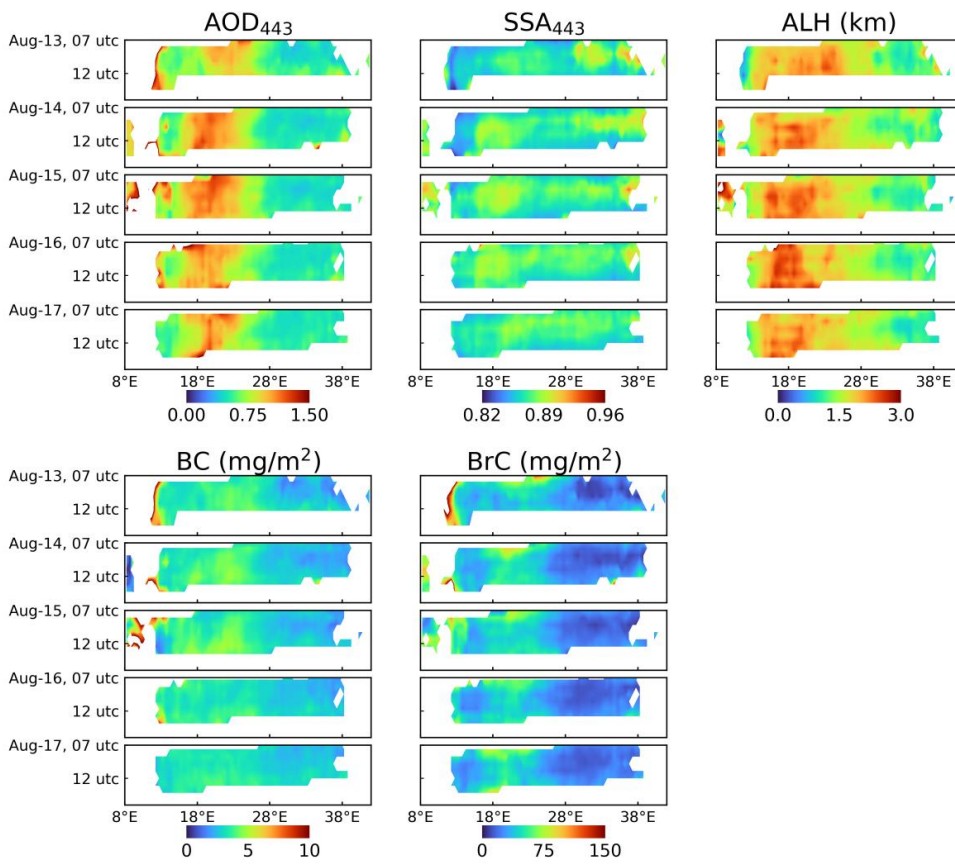

Figure 7. Same with Fig 5 except for over Central Africa (8-42°E, 17°S-5°N, 0.5° longitudinal
interval) from 13 to 17 August 2018.
The observed difference between the two regions clearly correlates with the different fuel
types – forests in North America and savannah grasses and bushes in Central Africa. For instance,



forest wildfires in North America with much higher thermal energy density result in elevated ALH,
incomplete combustion, and higher BrC concentrations, whereas fast-spreading grassland fires are
known for high BC concentration from flaming combustion emissions, but lower energy density,
which keep generated smoke generally within the boundary layer over Central Africa.

**3.2 Comparison of smoke properties derived from AERONET and CALIOP**

The regional validation of AOD, spectral SSA, and ALH throughout 2018 using the
AERONET and CALIOP datasets is presented in Fig 8. The AOD comparison over North America
demonstrates a correlation coefficient ($R$) of 0.91 and a root mean squared error ($rmse$) of 0.22. It
is important to note that this comparison only covers smoke retrievals; it excludes low AOD
conditions (e.g., background AOD at 443 nm < 0.4), that may result in lower validation statistics
compared to the previous analysis incorporating the combined "background+smoke" AOD ($R$ of
0.85 and $rmse$ of 0.13 in Lyapustin et al., 2021b). Nonetheless, the mean bias error ($MBE$) of 0.02
in version 3 is smaller than the 0.05 reported by Lyapustin et al. (2021b) based on v2. The fraction
within expected error ($EE\%$), defined as $\pm(0.05+0.2\times\text{AERONET AOD})$, is 74.9%. Central Africa
AOD also exhibits similar validation statistics, except for a lower $R$ (0.60), likely due to a narrow
range of collocated AOD compared to North America. However, the $MBE$ of −0.04 and EE% of
74.8% are comparable to the statistics for North America. Despite the absence of IR channels for
cloud detection and the relatively coarse spatial resolution (>10 km) of EPIC, which can lead to
sub-pixel cloud contamination (Marshak et al., 2018), the achieved accuracy in AOD retrieval is
very encouraging.
Regional comparisons of SSA with AERONET retrievals are more distinct than those of
AOD. Overall, the $SSA_{443}$ over North America from EPIC is lower than that from AERONET with
$MBE$ of −0.03 and EE fraction (within $\pm$ 0.03 + AERONET SSA; $EE_{0.03}\%$) of 45.2%. The
collocated range spans about 0.88 to 0.97 from EPIC and 0.90-1.00 from AERONET.
Comparisons over Central Africa show a much smaller bias ($MBE$ of −0.01) and higher $EE_{0.03}\%$
of 74.1%. The regional difference in accuracy could be attributed to uncertainty in our assumptions
of regional smoke model properties (e.g., particle size and real refractive index). Nonetheless, the
retrieved MAIAC EPIC $SSA_{443}$ remains comparable to OMAERUV $SSA_{440}$ retrievals ($rmse$ of
0.04 and $EE_{0.03}\%$ of 57.5% over North America; $rmse$ of 0.04 and $EE_{0.03}\%$ of 66.4% over South
America and Southern Africa in Jethva et al., 2014) and TropOMAER $SSA_{440}$ retrievals ($rmse$ of
0.04 to 0.04; $EE_{0.03}$ of 48 to 51% in Torres et al., 2020). Additionally, it is worth noting that the
current AERONET algorithm has a strong spectral smoothness constraint for the imaginary part
of refractive indices, resulting in less representation of BrC (Sinyuk et al., 2022; Eck et al., 2023).
By employing the relaxed constraint, they found decreased SSA (e.g., more absorbing) with
smaller sky radiance error from wildfire cases containing a large amount of BrC. However for the
biomass burning cases shown in Sinyuk et al. (2022) for both North America wildfire smoke and
savanna burning smoke in Zambia the difference in spectral SSA at 443 nm were ~0.01 or less for
the relaxed versus standard V3 constraints, while some differences in SSA at 675 nm were ~0.02
for North American smoke only. With this update from the AERONET side, we anticipate a





potentially better agreement between EPIC and AERONET for SSA$_{443}$ and possibly better for
SSA$_{680}$ in the future.
SSA$_{680}$ retrievals from North America show better agreement with AERONET than SSA$_{443}$
with a smaller *MBE* of −0.002, *rmse* of 0.02, and higher *EE$_{0.03}$%* (79.8%). However, Central Africa
shows slightly less agreement in SSA$_{680}$ compared to SSA$_{443}$, with a higher positive bias (*MBE* of
0.03) and smaller *EE$_{0.03}$%* of 60.2%. Additionally, the retrieved range of SSA$_{680}$ is relatively
narrower (~0.87 to 0.92) than that of AERONET (~0.80 to 0.99). Regardless, the statistics metrics
are much closer to POLDER GRASP SSA$_{680}$ retrievals (*rmse* of 0.06; *MBE* of −0.04 to −0.02 in
Chen et al., 2020).
The comparison of EPIC ALH with CALIOP also reveals strong regional dependence.
Most collocated ALH retrievals are relatively high over North America (3-4 km) and sometimes
reach 6-7 km. In Central Africa, ALH ranges from 0 to 4-5 km, with most collocated retrievals
falling within 1-3 km. The *rmse* value is closely related to the range of ALH; thus, it is relatively
high in North America (1.32 km). More favorable validation statistics were extracted from Central
Africa (*rmse* of 0.84 km; *EE$_{0.5km}$* of 49.4 %; *MBE* of −0.28 km). This level of accuracy, derived
from long-term validation rather than selected individual  cases, is better than the operational
TROPOMI ALH (*MBE* of −2.41 to −1.03 km and *rmse* of 1.97-3.56 km in Nanda et al., 2020).

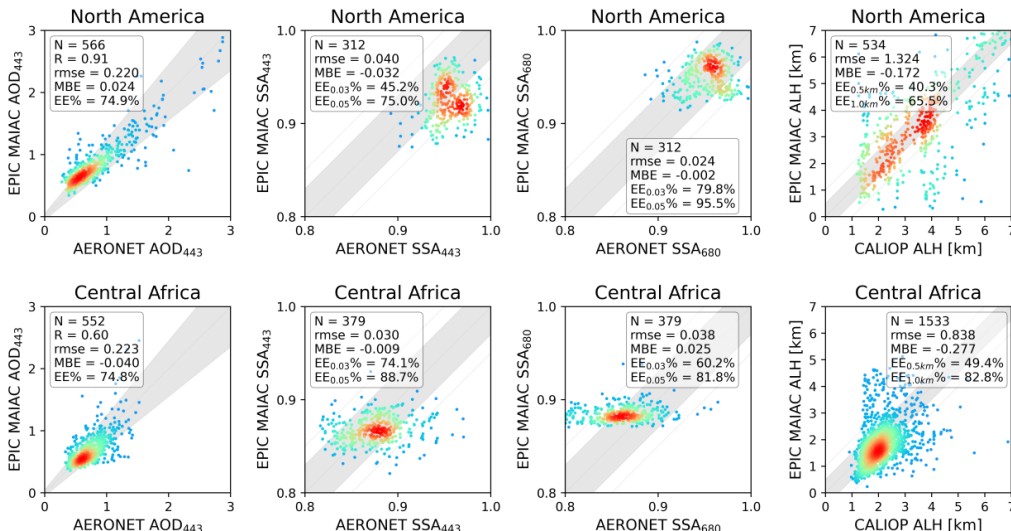

Figure 8. Comparison of EPIC smoke AOD$_{443}$ (first column), SSA$_{443}$ (second column), SSA$_{680}$
(third column) with AERONET, and ALH with CALIOP (fourth column). Color represents the
relative frequency of retrievals. The gray dashed lines and shaded areas are the 1:1 reference line
and ranges of expected error: ± (0.05 + 0.2 × AERONET AOD); ± (AERONET SSA + 0.03) or ±
(AERONET SSA + 0.05); and ± (CALIOP ALH + 0.5 km) or ± (CALIOP ALH + 1.0 km).





### 3.3 Regional climatology of smoke properties

We compiled all the smoke properties retrieved for 2018 and conducted a regional analysis to understand their climatology and relationships with environmental factors such as vegetation and fuel type, as well as meteorological conditions. Regional geographical distributions are illustrated in Fig 9, and the corresponding statistical distributions are presented as box-whisker plots in Fig 10. Regional- and monthly-averaged BC and BrC mass concentrations are presented in Fig 11.

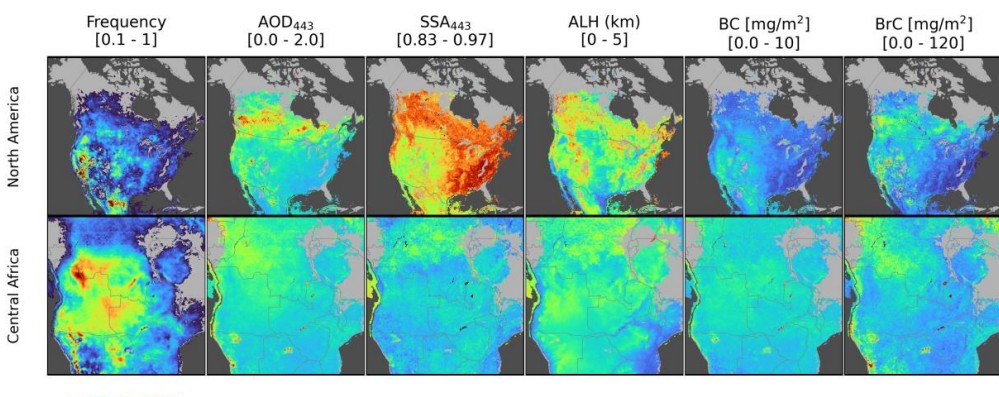

Figure 9. Spatial distribution of relative retrieval frequency (i.e., relative number of retrievals) and smoke properties (AOD$_{443}$, SSA$_{443}$, ALH, and BC and BrC mass concentrations) for 2018 over North America (top panels) and Central Africa (bottom panels). Pixels with retrieval frequencies lower than 10% compared to the regional maximum are filtered out. The color bar scale is indicated at the top of each panel.



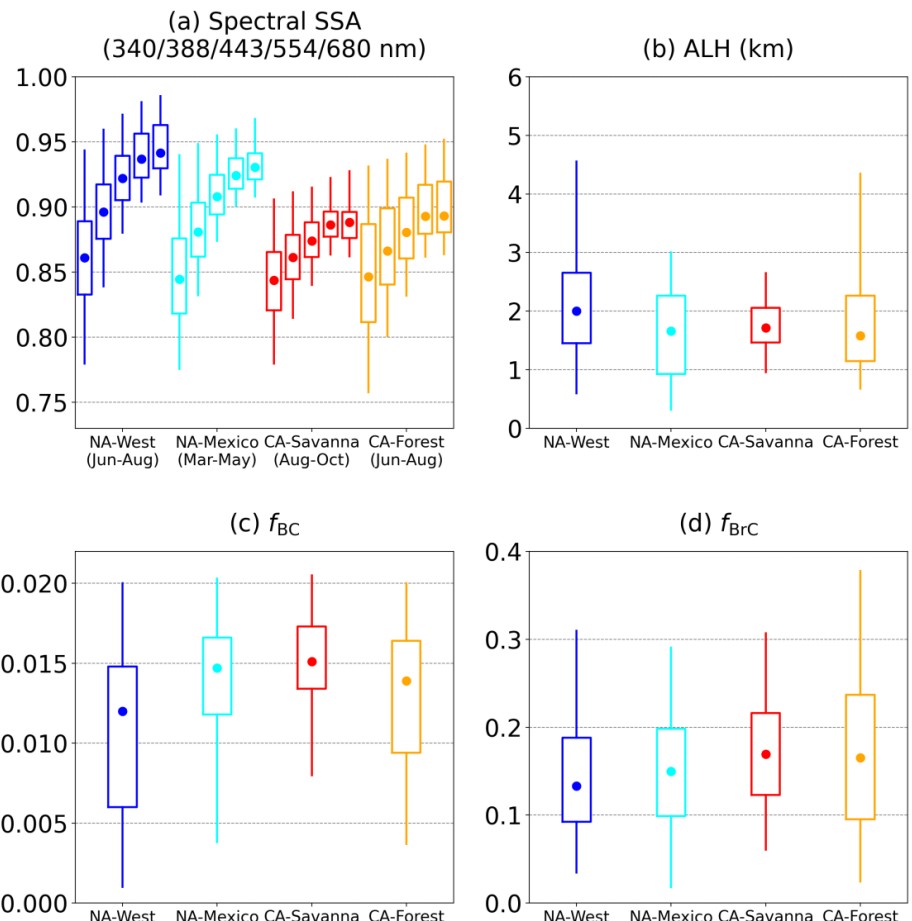

Figure 10. Distribution of (a) spectral SSA, (b) ALH, (c) BC volume fraction, and (d) BrC volume
fraction over western North America ("NA-West") and Mexico ("NA-Mexico") in North America,
and savanna ("CA-Savanna") and tropical forest ("CA-Forest") in Central Africa. Whiskers give
the 5th and 95th percentiles; boxes represent the 25 and 75th percentiles; and dots denote the 50th
percentile. In (a), five consecutive box-whisker plots for each region represent different
wavelengths (340, 388, 443, 554, and 680 nm from left to right).

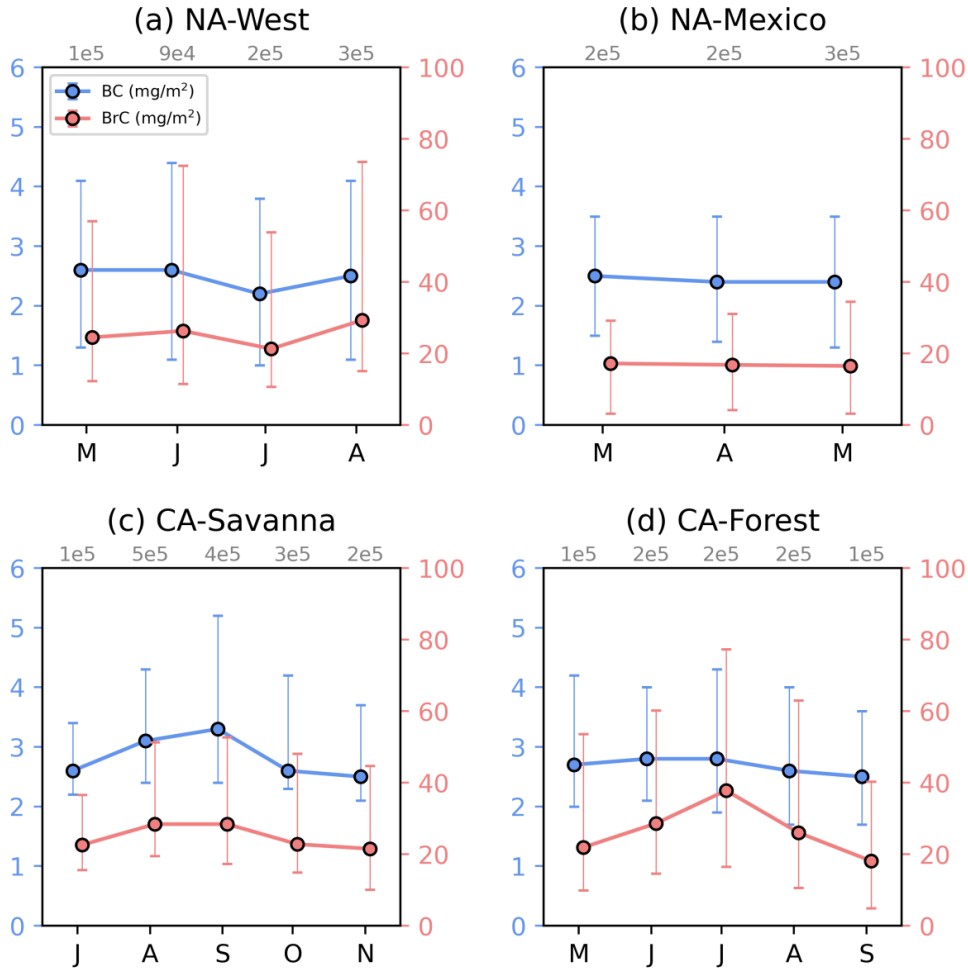

Figure 11. Regional monthly BC and BrC mass concentrations over western North America ("NA-West") and Mexico ("NA-Mexico") in North America, and savanna ("CA-Savanna") and tropical forest ("CA-Forest") regions in Central Africa. Whiskers denote the 15.9th and 84.1st percentiles; dots denote the 50th percentile. The number of smoke retrievals is displayed in grey at the top of each panel.

Active wildfires occur in late spring and summer over western North America, with expanded burned areas over the years (Dennison et al., 2014; Kalashnikova et al., 2018; Liu et al., 2010). Most smoke retrievals were detected over the western United States (e.g., California, Oregon, Washington) and western Canada (e.g., British Columbia) (Fig 9). The optical properties were quite distinct between source regions and downwind regions. The western US and western



Canada source regions show relatively low SSA and ALH, while central Canada, which is a source
region, but also mostly downwind regions for transported heavy smoke plume from western
regions, show higher SSA and ALH. This difference is closely related to the smoke aging process
discussed in Sec 3.1.1. Spatiotemporally integrated spectral SSA over western North America
("NA-West" region in Fig 2) of 0.86, 0.89, 0.92, 0.94, and 0.95 at 340, 388, 443, 554, and 680 nm,
respectively, align with the range 0.915-0.935 at 443 nm and 0.95-0.97 at 680 nm derived from
multiple AREONET measurements in September 2020 (Eck et al., 2023). The mean and standard
deviation of ALH was $2.2 \pm 1.2$ km with a wide range of values up to 4.6 km at the 95th percentile
(Fig 10b). The mean BC volume fraction of $0.011 \pm 0.006$ was the lowest among the selected
regions. The number of smoke pixels was maximum in August, with the highest BrC mass
concentration (median value of 29 mg/m$^2$), synchronized with seasonal wildfire activities over
western North America. Although BC and BrC concentrations can reach up to more than 5 mg/m$^2$
and 100 mg/m$^2$, respectively, over some specific regions (Fig 9), the averaged values were not as
high due to high spatiotemporal variation (Fig 11a). Another smoke-dominated region in North
America is found over Mexico ("NA-Mexico" region in Fig 2), where both natural wildfires and
agricultural burns occur annually during the hot and dry season (March to May; Rios et al., 2023).
This region exhibited smoke properties with more absorption and lower ALH with lower variation
($1.6 \pm 0.9$ km) than western US.
Central Africa is climatologically the largest global biomass burning source, peaking
during the austral winter. The region contributes approximately one-third of Earth's biomass
burning emissions from various sources, including wildfires, agricultural fires, and industrial
activities (van der Werf et al., 2010). The distribution of smoke retrievals appears relatively
homogeneous and similar to that of detected fires, with widespread retrieval frequency in Angola,
Democratic Republic of the Congo, and Zambia, and more varied sources in Namibia (Fig 9).
During the August–October burning season in Central Africa, aerosol light-absorption is
predominantly attributed to BC, a byproduct of savanna burning characterized by significant
flaming-phase combustion (Ward et al., 1996). Although the retrieved smoke AOD is not as high
as in North America, light absorption over savanna region in Central Africa ("CA-Savanna" region
in Fig 2) was more substantial, leading to higher BC and BrC mass concentrations. Low SSA
spanned from UV through the visible (0.84, 0.86, 0.88, 0.89, and 0.89 at 340, 388, 443, 554, and
680 nm, respectively), with higher BC and BrC volume fractions of 0.015 and 0.178, respectively.
The ALH is lower and less variance ($1.8 \pm 0.6$ km; 2.6 km for the 95th percentile) that of western
North America. The BC and BrC mass concentrations increased from July, peaked in September
(median values of 3.3 mg/m$^2$ and 28.4 mg/m$^2$, respectively), and declined toward November (Fig
11c); this aligns with long-term AERONET AOD measurements (Eck et al., 2003) and with
AERONET-based BC and BrC estimations (Schuster et al., 2016). By contrast, smoke from
tropical forest fires in Central Africa ("CA-Forest" region in Fig 2) shows slightly less absorption
with lower BC volume fraction (0.013) and larger variabilities of BrC volume fraction ($0.018 \pm$
0.11) and ALH ($1.9 \pm 1.1$ km) than that of savanna region. BC and BrC mass concentrations over





the tropical forest region in Central Africa peak in July (earlier than savanna region) with lower
BC (2.8 mg/m$^2$) and higher BrC (37.8 mg/m$^2$; Fig 11d) than those of the savanna region.
**4. Discussions**
**4.1 Comparison of the BrC to BC mass concentration ratio with other studies**

The ratio between OC and EC (OC/EC) is widely used to elucidate the apportionment of
carbonaceous components in smoke particles as a proxy for assessing the dominance of primary
emissions from flaming combustion (e.g., fossil fuel) versus smoldering combustion emissions
and secondary formation of OC (e.g., biomass burning, wildfires, secondary organic aerosol (SOA)
formation) (Lim and Turpin, 2002; Pokhrel et al., 2016). As BrC is an absorbing OC among total
OC, we inferred regional BrC-to-BC column mass concentration ratios (BrC/BC) from EPIC and
compared them with those from other studies providing BrC/BC or OC/EC.
Results of BrC/BC ratio from this study in North America and Central Africa are compared
with other previous studies in Fig 12. The absolute BC and BrC volume fractions in Central Africa
were higher than in North America, resulting in similar median values of the BrC/BC mass
concentration ratio (7.3 for North America and 8.0 for Central Africa). When the ratios are
categorized into different AOD ranges, the BrC/BC increases with AOD from both regions. For
two groups of AOD < 0.6 ("low-moderate AOD") and 0.6 < AOD < 2.0 ("high AOD"), the median
BrC/BC is higher in Central Africa (7.2 and 10.1) than in North America (6.9 and 8.9). The
variance, represented as the range of estimations, is more significant in North America for the two
groups, which could be ascribed to more diverse fuel types from natural, residential, and
agricultural sources and related emission processes (Xiong et al., 2022). For the cases of AOD >
2.0 ("extremely high AOD"), which corresponds to 2.6% and 0.7% of the entire retrieval record
in North America and Central Africa, respectively, North America showed a higher BrC/BC ratio
(median value of 41.5) with a higher variance than Central Africa (median value of 17.7). This
higher BrC/BC ratio in North America, compared to Central Africa, may have its origin in more
common smoldering combustion and/or more SOA formation during transport. Most "extremely
high AOD" cases were observed from transport plumes, where the increased BrC/BC ratio is
associated with their aging processes including SOA formation. These results are consistent with
POLDER/GRASP and MISR aerosol components analysis (Li et al., 2022; Junghenn Noyes et al.,
2022).
Our estimates exhibit relatively high variance because they encompassed all pixels detected
as smoke in the retrieval algorithm over the continents in 2018, rather than being limited to selected
heavy plumes. The national average of OC/EC ratio (3.6±0.9) obtained from U.S. EPA ground-
based chemical composition measurement networks (including CSN and IMPROVE) for all
sources, not only for smoke sources, (Cheng et al., 2024) falls within the estimates from EPIC's
"low-moderate AOD" group. OC/EC ratios obtained from specific wildfire samples including WE-
CAN campaign during 2018 July-September over western US (Liang et al., 2022; Carter et al.,
2021) range from approximately 14 to 100, corresponding to the "extremely high AOD" group. It



is important to note that although the BrC/BC ratio is smaller than the OC/EC ratio, obtaining an
accurate BrC/BC is challenging without proper measurements separating BrC from OC, which is
rarely done in experiments.
The ORACLES (August−September 2016) and CLARIFY (August 2017) campaigns over
the eastern South Atlantic Ocean (Carter et al., 2021) measured transported smoke aerosols from
Central Africa. The general level of AOD at 550 nm for both campaigns was ~0.3 to ~0.7
(Haywood et al., 2021; Sayer et al., 2019), and corresponding OC/EC ratios were 5-7, which are
consistent with the estimated EPIC ranges for "low-moderate AOD" and "high AOD". Another
comparison can be made with the BrC/BC mass concentration ratio inferred from AERONET
measurements (Schuster et al., 2016). Although the definition is similar to ours, both using
column-integrated and remote-sensing-based values, it shows relatively lower values than ours.
This difference could be attributable to the different wavelengths (i.e., UV-Vis for EPIC, Vis-NIR
for AERONET) used for the measurements and different assumptions in the components (e.g.,
dependence of composition on particle size in Schuster et al., 2016).
The EPIC BrC/BC ratios increased with AOD, representing aging processes during
transport over North America and Central Africa. They are generally consistent with other studies
despite different measurement characteristics, such as OC/EC vs. BrC/BC, and *in-situ* versus
remote sensing.

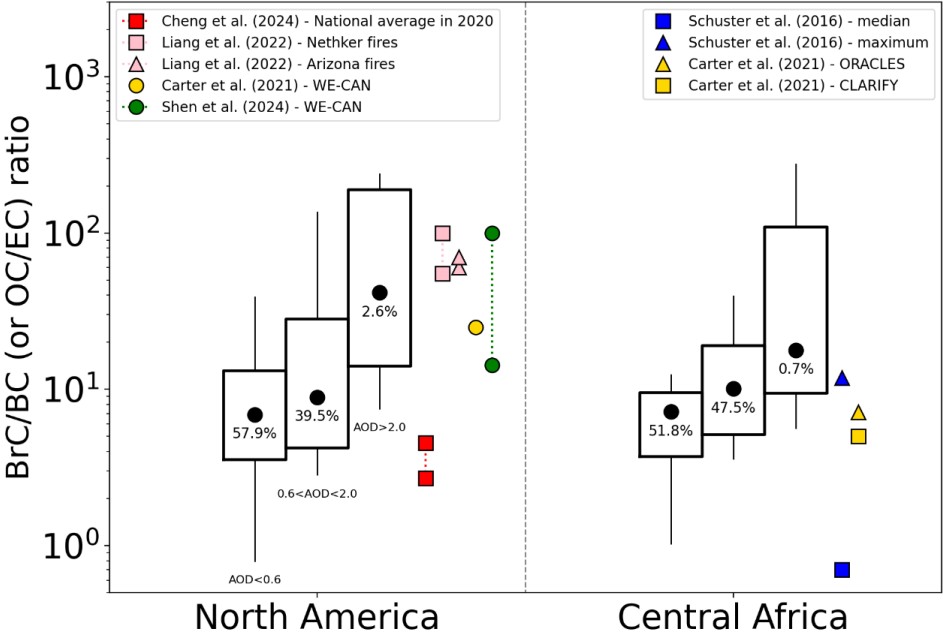




Fig 12. Regional EPIC-derived BrC to BC column mass concentration ratios across three AOD ranges (AOD < 0.6, 0.6 < AOD < 2.0, and AOD > 2.0). Each box-whisker plot comprises the 5th, 15.9th, 50th, 84.1st, and 95th percentiles. The percentages of retrievals per each AOD range are denoted within the box. On the right side of each region panel, the values (or range) of the BrC-to-BC ratio (only for Schuster et al., 2016) or OC-to-BC ratio (all others) from other studies are shown.

## 4.2 Uncertainty of volume fractions due to assumed BC and BrC refractive indices

Assumed spectral imaginary refractive indices of BC and BrC determine their inferred volume fractions. Identical spectral absorption can result in lower BC and BrC fractions with higher BC and BrC imaginary refractive indices and vice versa. As most satellite measurements, including EPIC, lack sensitivity to infer both the imaginary refractive indices of inclusions and their volume fractions, we must assume the imaginary refractive indices of inclusions to infer their volume fractions. Here, we investigate the effect of this assumption on the inferred volume fractions and assess the resulting uncertainties.

A total of three different BC datasets were considered (Fig 13a). "BC1", which we used, and "BC2" were derived from multiple measurements combined with the assumption that light-absorbing carbon has a single refractive index and that variation can be expressed by the Bruggeman effective-medium theory (Bond and Bergstrom, 2006). "BC3", utilized in aerosol modeling for AirMSPI analysis (Kalashnikova et al., 2018), was originally referred to as the "soot" component of the Optical Properties of Aerosols and Clouds (OPAC) dataset described in Hess et al. (1998). The value of k is between 0.4 and 0.8 and is spectrally invariant or nearly invariant.

We tested nine different BrC datasets (Fig 13b). "BrC1", which we used, was derived from organic carbon extracted from wood burning and SAFARI biomass smoke samples as described in Kirchstetter et al. (2004). "BrC2" is an Air-MSPI retrieved value during the FIREX-AQ campaign (O. Kalashnikova, personal communication, May 19, 2020). "BrC3" represents aerosols emitted from the smoldering combustion of Boreal and Indonesian peatlands (Sumlin et al., 2018). "BrC4", "BrC5", and "BrC6" represent water-insoluble BrC with relative humidity of 0%, 75%, and 99%, respectively, calculated by combining the upper curve of Sun et al. (2007) and hygroscopic properties in Rissler et al. (2006). "BrC7", "BrC8", and "BrC9" are the same but represent water-soluble BrC. These datasets were obtained from the Table of Aerosol Optics (TAO) dataset within the framework of the Models, In situ, and Remote sensing of Aerosols (MIRA) working group projects (https://science.larc.nasa.gov/mira-wg/).

Here, two smoke cases were analyzed: "Case 1" ($k_0$ of 0.007 and SAE of 2) and "Case 2" ($k_0$ of 0.012 and SAE of 1.5), representing the most populated EPIC retrievals in the AERONET validation over North America and Central Africa, respectively. For Case 1, the $f_{BC}$ and $f_{BrC}$ based on our current assumptions are 0.011 and 0.112, respectively (marked with a "star" marker in dark blue in Fig 13c). With different assumptions for inclusion properties, they have a range of 0.008-0.031 and 0.096-0.982, respectively. Less absorbing BC assumptions (i.e., smaller $k$) result in increased $f_{BC}$ to 0.012 ("BC2" and "BrC1") and 0.018 ("BC3" and "BrC2"). The maximum





difference of $f_{BC}$ is 0.013, with the lowest absorption in BrC ("BrC9"). The potential $f_{BrC}$ values
exhibit greater variability. The $f_{BrC}$ value with the current assumption (0.112) is one of the lowest
values among tested combinations and similar to those from "BrC2" and "BrC3", which have
stronger absorption than others. The BrC assumptions with less absorbing properties show higher
$f_{BrC}$ from 0.264 to 0.981. We also tested the spectral $k$ for dark BrC obtained from the FIREX-AQ
campaign in the western US (Chakrabarty et al., 2023). They showed an estimated $f_{BC}$ close to
zero because of the relatively high $k$ of 0.1 at 680 nm. Case 2 is converted to higher $f_{BC}$ (0.019)
and similar $f_{BrC}$ (0.117) compared to Case 1 with the default assumption. The range of $f_{BC}$ and
$f_{BrC}$ from the different combinations is 0.016-0.047 and 0.101-0.980, respectively. It is essential
to acknowledge that inferring volume fractions and mass concentrations is based on assumed
inclusion properties, introducing some uncertainties. The assumed properties of BC and BrC will
need to be refined in future studies (e.g., a suggested concept in Kahn et al., 2017) to enhance the
accuracy of our findings.

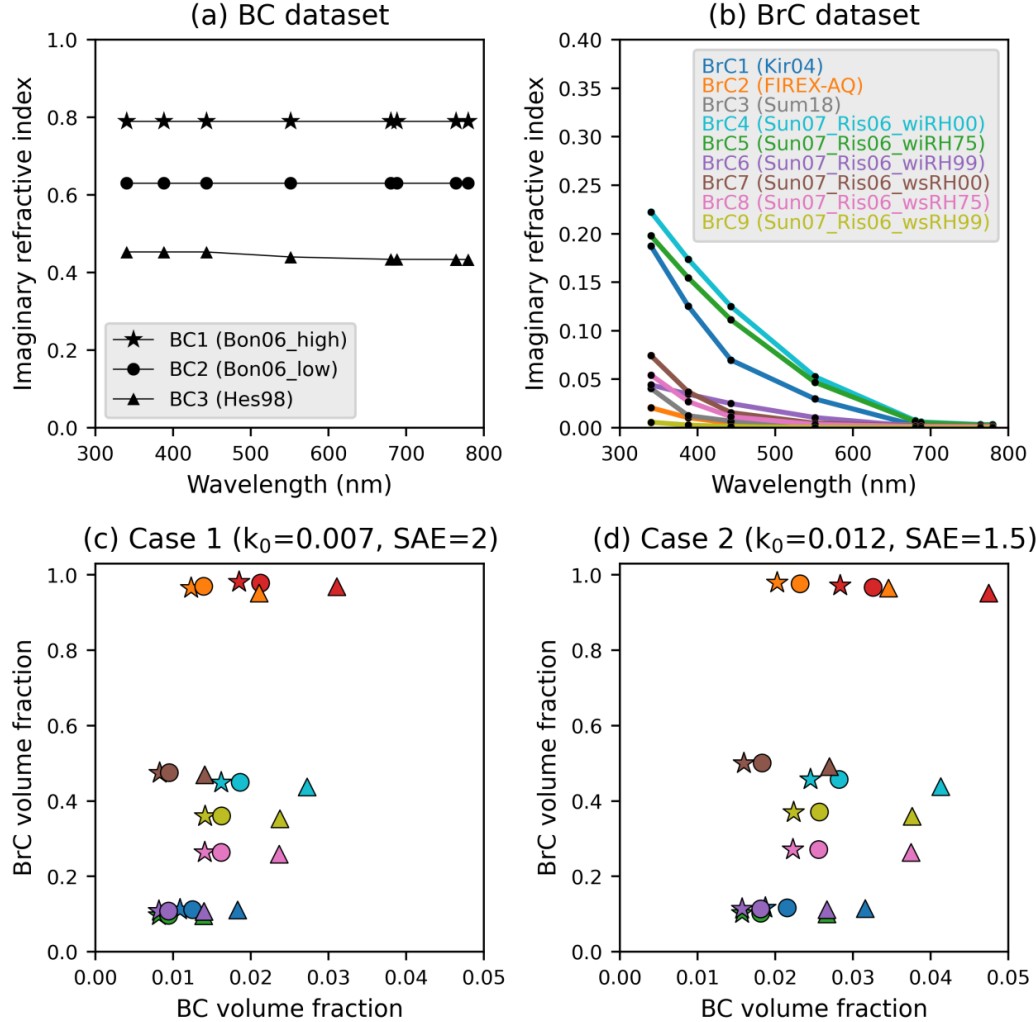

Fig 13. Spectral imaginary refractive indices of (a) BC and (b) BrC. Range of BC and BrC volume fractions for (c) "Case 1" ($k_0$ of 0.007 and SAE of 2) and (d) "Case 2" ($k_0$ of 0.012 and SAE of 1.5). The star, circle, and triangle symbols in (c) and (d) refer to different BC assumptions of "BC1", "BC2", and "BC3" in (a). The different colors in (c) and (d) refer to different BrC assumptions from "BrC1" to "BrC9" in (b).

## 5. Summary and conclusions

This study introduced a technique inferring the BC and BrC light-absorbing components of smoke aerosol by leveraging the spectral absorption retrieved in the MAIAC EPIC algorithm. Spectral absorption retrievals allowed us to quantify the BC and BrC fractions, which were then



converted to column-integrated mass concentrations assuming the particle mass extinction
efficiency. We assumed that BC and BrC are internally mixed with a non-absorbing host
representing non-absorbing OC, sulfate, nitrate, or ammonium components, using the Maxwell
Garnett effective medium approximation.

We analyzed regional characteristics over North America and Central Africa in 2018,
utilizing all available MAIAC EPIC smoke property retrievals (AOD, spectral SSA, ALH, and BC
and BrC volume fractions and mass concentrations). Selected cases showed that smoke aerosols
emitted from wildfires over western North America exhibited extremely high AOD up to ~6 with
elevated ALH (6-7 km). Dynamic changes in spectral absorption and significant BrC components
were observed during continental-scale transport. The EPIC MAIAC products successfully
monitored the transport and evolution of smoke optical properties with high temporal resolution
during regional-to-continental-scale transport. Biomass-burning smoke over Central Africa
displayed higher absorption with greater BC and BrC fractions than North America, showing
seasonal changes in major source locations. They also showed less strong zonal transport with
ALH closer to the surface, and diurnal change in smoke amounts related to fire activities.

EPIC-retrieved $AOD_{443}$, $SSA_{443}$, $SSA_{680}$, and ALH agreed with collocated AERONET and
CALIOP measurements with *rmse* of 0.2, 0.03-0.04, 0.02-0.04, and 0.8-1.3 km, respectively, and
the overall accuracies were comparable to other operational satellite products such as OMI,
TROPOMI, and POLDER. Spatiotemporally integration of measurements revealed geographical
characteristics and distinct differences in optical properties, ALH, and inferred BC and BrC,
closely linked to burning types and meteorological conditions. Smoke from forest fires in western
North America shows $SSA_{443}$ of 0.92 with low BC volume fraction of 0.011 and high ALH with
larger standard deviation (2.2 ± 1.2 km). The wildfires and agricultural fires over the Mexico
region generated smoke with more absorption and lower ALH. The Savanna region in Central
Africa during August to October shows smoke properties with most absorbing with high BC and
BrC volume fractions (0.015 and 0.178, respectively) and lower ALH with smaller variation.
Smoke from tropical forests in Central Africa exhibits absorption between that of western US and
savanna regions and high ALH variability. The impact of assumed imaginary refractive indices of
BC and BrC in estimating their volume fractions was analyzed based on a literature survey,
presenting the corresponding uncertainty ranges of our retrievals.

Although we focused on North America and Central Africa, smoke aerosols have a
significant impact on air quality and climate globally. Future studies will extend the analysis to
other regions using almost a decade of EPIC measurements since 2015, with extensive validation
and error analysis using multiple measurements, including AERONET, CALIOP, and in-situ
aerosol composition data.

The MAIAC EPIC smoke aerosol components presented here could serve as valuable *a*
*priori* information for recent and upcoming satellite missions such as the Plankton, Aerosol, Cloud,
ocean Ecosystem (PACE; https://pace.gsfc.nasa.gov/) (Remer et al., 2019a, b), the Multi-Angle
Imager for Aerosols (MAIA; https://maia.jpl.nasa.gov/) (Diner et al., 2018), EPS-SG Multi-
Viewing Multi-Channel Multi-Polarisation Imaging (3MI) (Fougnie et al., 2018) and Atmosphere



Observing System (AOS; https://aos.gsfc.nasa.gov/), focusing on retrieving aerosol microphysical
and optical properties, and inferring chemical composition, with higher accuracy from multi-angle
polarization measurements. Integration of our results with other in-situ and remote sensing
measurements and models (e.g., Kahn et al., 2023) should enhance our understanding of smoke
aerosol aging processes, improve air quality monitoring and forecasting, and refine the
quantification of radiative forcing due to smoke aerosols on a global scale.
**Author contributions**
M. Choi and AL designed the study with discussions with GLS and SG. GLS provided
major guidance on developing the BC and BrC estimation algorithms. AL and WY provided the
MAIAC EPIC products. AL and SK conducted RT calculations (LUTs for MAIAC). M. Choi, AL,
YW and SG developed the code and performed the retrievals. GLS and OK participated in the
collection of refractive indices data. M. Choi, AL, GLS, and SG analyzed the results. M. Choi and
AL wrote the manuscript with comments from all co-authors.
**Competing interests**
The authors declare that they have no conflict of interest.
**Data availability**
The retrievals can be requested directly from the corresponding author
(myungje.choi@nasa.gov or alexei.i.lyapustin@nasa.gov).
**Acknowledgment**
The work of A. Lyapustin, M. Choi, S. Go, and Y. Wang was funded by the NASA
DSCOVR program (21-DSCOVR-21-0004; manager Dr. R. Eckman) and in part by the NASA
PACE program (19-PACESAT19-0039). J. S Reid was funded by the Office of Naval Research,
Code 322. The work of H. Moosmüller was supported in part by the National Science Foundation
under Grant No. OIA- 2148788 and by NASA under grant 80NSSC20M0205 (PACE SAT Project:
PACE UV ROAD). We are grateful to the AERONET team for providing validation data and to
the NASA Center for Climate Simulations providing resources for the EPIC data processing.

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
