# Peer review of "Light-absorbing black carbon and brown carbon components of smoke aerosol from"

_EGUsphere, 2024_

## Author Response (AR1)

**Dear Anonymous Referee #1**

**We appreciate the insightful questions and comments from the reviewers. After careful consideration, we have incorporated a detailed item-by-item response for your comments. The reviewer's comments are in plain font. Our responses are in bold font.**

Comment: This paper uses observations from EPIC instrument onboard the DSCOVR spacecraft to study aerosol plumes generated by biomass burning over North America and central Africa with a focus on their light absorption properties. The paper is very clear, it uses an original dataset and focus on a process that can be of interest for a wide community. It could then be published with little change. I feel nevertheless that some of the figure contain little information and could therefore be removed.
Figure 5 and 7 show Hovmöller diagrams of various aerosol parameters. The physical interpretation of these diagrams is unclear. It seems difficult to interpret these figure as an evolution of the aerosol plume during transport as one can observe an increase of the optical depth. Also, the direction of the transport is not fully clear. Since little interpretation is made from these figure, I recommend to remove them.

**Response: To avoid uncertainty regarding "evolution" or "direction of the transport," we changed the term "change of smoke optical properties during transport" to "regional-to-continental scale variability of smoke optical properties." We retained the Hovmöller diagrams for North America and Central Africa because they effectively showcase EPIC's unique capability for high-temporal resolution measurements over global regions, a feature not available from any other current or past single LEO or GEO sensors. These diagrams also highlight the differences in the spatiotemporal distribution between the two regions.**

Comment: Similarly, I am not sure how figure 9, that shows mean parameters over a full year can be interpreted. The averages put together situations that are very different, with days that are affected by biomass burning plumes and other that are not. As a consequence, I wonder what sense what can make from the mean SSA values (or the aerosol height).

**Response: The presented climatology of MAIAC EPIC smoke properties and the relative retrieval frequency provide spatial features of smoke optical properties for moderate and high aerosol loading cases. Although the spatial distribution is generated from the entire annual samples, the data is highly concentrated in the smoke-dominant season (as indicated by the grey numbers at the top of each panel in Fig 11 of the preprint). Fig 9 provides a detailed spatial distribution from regional (e.g., within the subregions described in Fig 2) to continental scales. North America exhibited a relatively higher SSA at 443 nm (>0.9) with more significant spatial variability (e.g., West vs. East) compared to Central Africa, which showed a relatively lower SSA (<0.9) with less spatial variability across burning regions, complementing Fig 10. These spatial characteristics can benefit climatological studies, such as evaluating climate models.**

Comment: Finally, Figure 11 shows 4 plots of the monthly variations of the BC and BrC column concentrations. There is no significant monthly variations to show. As a consequence, I think that a single sentence "there is no significant temporal variation in the monthly mean values" would be sufficient to carry the message.

**Response: Fig 11 has been removed, but the corresponding explanations have been retained in the text.**

Other comment

There seem to be some error in Figure 8 or its legend. It is hard to see a "gray dashed line"

**Response: Fig 8 has been updated to improve readability and to correct errors.**

Comment: There are some lines on the SSA plots the meaning of which are unclear. For the SSA, the expected error is 0.03 or 0.05, not Aeronet SSA+0.03 (ie the error does not vary with the value, contrarily to that of the AOD)

**Response: The defined expected error envelopes of SSA, AERONET SSA ± 0.03 or ± 0.05, are a validation statistic used to evaluate MAIAC EPIC SSA retrieval accuracy against AERONET SSA. This is a standard validation method for evaluating satellite SSA, as recommended by the Global Climate Observing System (GCOS) of the World Meteorological Organization (WMO, 2011). Assuming the AERONET SSA value is close to the "true value", the expected error range should be centered around the AERONET SSA. This method is widely used for validating satellite SSA products (e.g., Jethva et al., 2014; Lyapustin et al., 2021; Go et al., 2020).**

**References**

**Go, S., Kim, J., Mok, J., Irie, H., Yoon, J., Torres, O., Krotkov, N. A., Labow, G., Kim, M., Koo, J.-H., Choi, M., & Lim, H. (2020). Ground-based retrievals of aerosol column absorption in the UV spectral region and their implications for GEMS measurements. Remote Sensing of Environment, 111759. https://doi.org/10.1016/j.rse.2020.111759**

**Jethva, H., Torres, O., & Ahn, C. (2014). Global assessment of OMI aerosol single-scattering albedo using ground-based AERONET inversion. Journal of Geophysical Research, 119(14), 9020–9040. https://doi.org/10.1002/2014JD021672**

**Lyapustin, A., Go, S., Korkin, S., Wang, Y., Torres, O., Jethva, H., & Marshak, A. (2021). Retrievals of Aerosol Optical Depth and Spectral Absorption From DSCOVR EPIC. Frontiers in Remote Sensing, 2(March), 1–14. https://doi.org/10.3389/frsen.2021.645794**

**WMO (2011). Systematic observation requirements for satellite-based data products for climate, 2011 Update Supplemental details to the satellite-based component of the "Implementation Plan for the Global Observing System for Climate in Support of the UNFCCC", Tech. Rep., WMO, Geneva, Switzerland, https://library.wmo.int/idurl/4/48411**

Dear Anonymous Referee #2

We appreciate the insightful questions and comments from the reviewers. After careful consideration, we have incorporated a detailed item-by-item response for your comments. The reviewer's comments are in plain font.  Our responses are in bold font.

Comment: What made you set maximum AOD value to 6?

**Response: The maximum value of the lookup table AOD is set at 6 to cover most heavy aerosol loading cases while maintaining retrieval accuracy. This value is similar to that used in AERONET direct Sun measurements, which assumes the maximum (AOD × m) < 7, where m is the secant of the zenith angle for angles < 70° (Eck et al., 2019).**

Comment: AOD is better correlated to BrC in northern America (see in Fig 5), while it shows a better correlation in central Africa (see in fig 7). why is that? Or why do BC not correlate well with AOD in NA?

**Response: The BC and BrC properties presented in Figs 5 and 7 are mass concentrations, derived from the combined quantity of AOD and the volume fraction of BC or BrC with assumed densities. Since North America exhibited a relatively lower BC volume fraction than the Central Africa region (Fig 10), the BC concentration would be less correlated to AOD in North America than in Central Africa. The higher BC volume fraction in Central Africa can be attributed to savanna burning, characterized by significant flaming-phase combustion resulting in incomplete burning. In contrast, the lower BC volume fraction over North America is likely due to more common smoldering combustion from boreal forest fires.**

Comment: Do you think ALH correlates with convection due to the high surface temperature in Africa? How do fires in NA have more thermal energy for higher ALH?

**Response: The main reason is the difference in fuel type – forests in North America vs. grasses/bushes in Central Africa. High fuel consumption can explain higher ALH from North America with more thermal energy. Fuel consumption is defined as the amount of biomass, coarse and fine litter, and soil organic matter consumed per unit area burned. It is the product of fuel load and combustion completeness, leading to regional differences. For instance, western US, Canada, and Siberia regions categorized as boreal forests exhibit high fuel consumption (e.g., > 2 kg C m−2 burned), whereas the savanna region in Central Africa has lower fuel consumption (e.g., 1−2 kg C m−2 burned; van der Werf et al., 2017). The energy released along the flame front is directly related to plume height, with plumes from these fires reaching altitudes between 2.2 km and 13 km (Lavoue et al., 2020). Satellite-derived fire radiative power also shows significant differences between smoke plumes in the free troposphere (1620−1640 MW) and those within the boundary layer (174−465 MW; van der Werf et al., 2010). This response is added to Section 3.1.2 of the revised manuscript.**

Comment: How do the size distributions look between NA and Africa during fires

**Response: According to AERONET observations, the accumulation mode (radius <0.5 μm) size distributions over central/southern Africa are relatively consistent, with little difference in the peak modal radius (0.145−0.155 μm) and a geometric standard deviation of 1.55. In contrast, the coarse mode exhibits substantial differences in size and width of the mode (Eck et al., 2003). The size distribution of smoke aerosols from extreme forest fires over North America varies widely, with fine mode volume median radius ranging from 0.10 to 0.35 μm. The very large fine-mode particle radii (>0.25 μm) could result from a combination of**

fuel type, combustion phase, and aging processes (Eck et al., 2023). The regional differences in particle size distribution introduce uncertainty in our EPIC MAIAC algorithm, which assumes a single particle size distribution.

Comment: What impact do you think an externally mixed assumption would have made?

Response: External mixing of BC and BrC generally exhibits less absorption than internal mixing. A theoretical study by Lesins et al. (2002) demonstrated that external mixing of BC with ammonium sulfate, considered as the host in our study, yields higher single scattering albedo (i.e., lower absorption) than internal mixing by 3-12%, depending on different internal mixing assumptions. Optical measurements of biomass burning particles from the Four Mile Canyon fire near Boulder, Colorado, also showed that internal mixtures of BC and particulate organic matter enhanced absorption by up to 70% (Lack et al., 2012).

The topic of BC and BrC mixing is complex and multifaceted. BC is generally considered internally mixed with other aerosols, as summarized in Lesins et al. (2002). When generated by fossil fuel combustion, BC coexists with sulfates and other gases and particles. When produced by biomass burning, it coexists with organic materials. The hydrophobicity of BC is believed to decrease over time as its surface is attacked by oxidants in the atmosphere, making it more likely to appear in solution aerosols or cloud droplets. Schwarz et al. (2008) found that the internal mixing fraction of BC particles varies greatly, from 70% in fresh biomass burning plumes to 46% in the background and 9% in fresh urban emissions. For these reasons, recent remote sensing studies estimating BC and BrC focusing on wildfires and biomass burning have assumed internal mixing of components (e.g., Schuster et al., 2016; Li et al., 2019). However, quantifying errors associated with the mixing status remains challenging.

The following sentence is added in Sec 2.2 of the revised manuscript:
"External mixing could be assumed, resulting in lower absorption than internal mixing (Lesins et al., 2002; Lack et al., 2012), but most BC particles exist internally mixed with other components in biomass burning plumes (Schwarz et al., 2008)."

Comment: Did you also take the non-absorbing components into AOD,SSA,ALH etc calculation?

Response: Yes, we considered non-absorbing components in the retrieval process. In the step of AOD, spectral absorption, and ALH retrieval, the assumed smoke aerosol model varies in spectral absorption exponent (SAE) and imaginary refractive index at 680 nm (k680), as described in the manuscript. Lower values of SAE and k680 correspond to less absorbing or non-absorbing smoke aerosols.

For the retrieval of BC and BrC, spectral absorption is attributed to three different components: two absorbing inclusions (BC and BrC) within a host, under an internal mixing assumption using the Maxwell-Garnett medium approximation. The host represents non-absorbing components such as sulfate, nitrate, ammonium, or non-absorbing organic carbon (OC).

References

Eck, T. F., Holben, B. N., Ward, D. E., Mukelabai, M. M., Dubovik, O., Smirnov, A., Schafer, J. S., Hsu, N. C., Piketh, S. J., Queface, A., le Roux, J., Swap, R. J., & Slutsker, I. (2003). Variability of biomass burning aerosol optical characteristics in southern Africa during the SAFARI 2000 dry season campaign and a comparison of

single scattering albedo estimates from radiometric measurements. Journal of Geophysical Research: Atmospheres, 108(13). https://doi.org/10.1029/2002jd002321

Eck, T. F., Holben, B. N., Giles, D. M., Slutsker, I., Sinyuk, A., Schafer, J. S., Smirnov, A., Sorokin, M., Reid, J. S., Sayer, A. M., Hsu, N. C., Shi, Y. R., Levy, R. C., Lyapustin, A., Rahman, M. A., Liew, S. C., Salinas Cortijo, S. v., Li, T., Kalbermatter, D., … Aldrian, E. (2019). AERONET Remotely Sensed Measurements and Retrievals of Biomass Burning Aerosol Optical Properties During the 2015 Indonesian Burning Season. Journal of Geophysical Research: Atmospheres, 124(8), 4722–4740. https://doi.org/10.1029/2018JD030182

Eck, T. F., Holben, B. N., Reid, J. S., Sinyuk, A., Giles, D. M., Arola, A., Slutsker, I., Schafer, J. S., Sorokin, M. G., Smirnov, A., LaRosa, A. D., Kraft, J., Reid, E. A., O'Neill, N. T., Welton, E. J., & Menendez, A. R. (2023). The extreme forest fires in California/Oregon in 2020: Aerosol optical and physical properties and comparisons of aged versus fresh smoke. Atmospheric Environment, 305. https://doi.org/10.1016/j.atmosenv.2023.119798

Lack, D. A., Langridge, J. M., Bahreini, R., Cappa, C. D., Middlebrook, A. M., & Schwarz, J. P. (2012). Brown carbon and internal mixing in biomass burning particles. Proceedings of the National Academy of Sciences, 109(37), 14802–14807. https://doi.org/10.1073/pnas.1206575109

Lavoué, D., Liousse, C., Cachier, H., Stocks, B. J., & Goldammer, J. G. (2000). Modeling of carbonaceous particles emitted by boreal and temperate wildfires at northern latitudes. Journal of Geophysical Research: Atmospheres, 105(D22), 26871–26890. https://doi.org/10.1029/2000JD900180

Lesins, G., Chylek, P., & Lohmann, U. (2002). A study of internal and external mixing scenarios and its effect on aerosol optical properties and direct radiative forcing. Journal of Geophysical Research: Atmospheres, 107(9–10). https://doi.org/10.1029/2001jd000973

Li, L., Dubovik, O., Derimian, Y., Schuster, G. L., Lapyonok, T., Litvinov, P., Ducos, F., Fuertes, D., Chen, C., Li, Z., Lopatin, A., Torres, B., & Che, H. (2019). Retrieval of aerosol components directly from satellite and ground-based measurements. Atmospheric Chemistry and Physics, 19(21), 13409–13443. https://doi.org/10.5194/acp-19-13409-2019

Schuster, G. L., Dubovik, O., & Arola, A. (2016). Remote sensing of soot carbon – Part 1: Distinguishing different absorbing aerosol species. Atmospheric Chemistry and Physics, 16(3), 1565–1585. https://doi.org/10.5194/acp-16-1565-2016

Schwarz, J. P., Gao, R. S., Spackman, J. R., Watts, L. A., Thomson, D. S., Fahey, D. W., Ryerson, T. B., Peischl, J., Holloway, J. S., Trainer, M., Frost, G. J., Baynard, T., Lack, D. A., de Gouw, J. A., Warneke, C., & del Negro, L. A. (2008). Measurement of the mixing state, mass, and optical size of individual black carbon particles in urban and biomass burning emissions. Geophysical Research Letters, 35(13). https://doi.org/10.1029/2008GL033968

van der Werf, G. R., Randerson, J. T., Giglio, L., Collatz, G. J., Mu, M., Kasibhatla, P. S., Morton, D. C., DeFries, R. S., Jin, Y., & van Leeuwen, T. T. (2010). Global fire emissions and the contribution of deforestation, savanna, forest, agricultural, and peat fires (1997–2009). Atmospheric Chemistry and Physics, 10(23), 11707–11735. https://doi.org/10.5194/acp-10-11707-2010

van der Werf, G. R., Randerson, J. T., Giglio, L., van Leeuwen, T. T., Chen, Y., Rogers, B. M., Mu, M., van Marle, M. J. E., Morton, D. C., Collatz, G. J., Yokelson, R. J., & Kasibhatla, P. S. (2017). Global fire emissions estimates during 1997–2016. Earth System Science Data, 9(2), 697–720. https://doi.org/10.5194/essd-9-697-2017